# CRAMER: Control via Request-Aware Masking for Editing Recommenders

**Zhiyuan Julian Su** [1 2 †]   **Naihe Feng** [2]   **Zhen Luther Qin** [2]   **Ga Wu** [2]

## Abstract

Sequential recommendation models, while powerful, have limited flexibility in responding to immediate user requests, making it difficult to adapt their recommendations to the user's timely interests. Unfortunately, existing user request adaptation methods often incur high computational overhead due to either 1) retraining the entire backbone network or 2) leveraging the inference ability of large language models (a.k.a. prompt engineering), limiting their applicability in large-scale recommendation services. This paper presents **C**ontrol via **R**equest-**A**ware **M**asking for **E**diting **R**ecommenders (**CRAMER**), a framework that takes users' natural-language requests to immediately change sequential recommendation models' behavior. Specifically, inspired by the model control theory, CRAMER treats user requests as control signals to modulate frozen sequential recommender backbone parameters through masking, achieving instant adaptation to diverse requests while avoiding costly retraining. Experiments on multiple large-scale benchmark datasets show that CRAMER outperforms four state-of-the-art request-aware baselines across multiple recommendation metrics while achieving minimal overhead. Moreover, the proposed framework exhibits enhanced controllability and cross-domain adaptability, establishing a new paradigm for request-aware sequential recommendation.

## 1. Introduction

Sequential recommendation models have advanced the state-of-the-art in predicting users' next interactions by modeling temporal patterns in behavior, but they remain inflexible when users issue immediate requests (Li et al., 2023;

Ye et al., 2025; Shen et al., 2026). Real-world users frequently express on-the-fly intents in response to an initial recommendation list (e.g., "I want more exciting games"), rather than issuing explicit search queries. Handling such *reactive, post-recommendation* feedback introduces challenges beyond conventional sequential recommendation. First, natural-language requests may emphasize or even contradict historical preferences, requiring the model to dynamically balance immediate intent with long-term behavior patterns (Gao et al., 2021; Radlinski et al., 2022; Jin et al., 2022; Chen et al., 2023). Second, requests often contain rich semantics such as negations, constraints, or fine-grained attribute preferences, which require accurate interpretation and controllable adjustment of recommendations (Jannach et al., 2021; Moradizeyveh, 2022). Finally, real-time adaptation must be achieved efficiently: sequential recommendation backbones are inherently complex, trained and deployed models, so request-aware extensions must rely on lightweight, parameter-efficient mechanisms that preserve responsiveness without full fine-tuning (Houlsby et al., 2019; Prottasha et al., 2024; Shao et al., 2025).

To address these gaps, existing research has primarily viewed request adaptation as either an input-level or output-level problem. Input-level approaches augment historical sequences with control tokens or vectors (He et al., 2022; Li et al., 2023), while output-level strategies employ re-ranking or filtering logic after the model has generated candidates (Liu et al., 2024; Liao et al., 2026; Zhang et al., 2025). Consequently, these methods face a dilemma: they either rely on shallow representations that cannot capture nuanced user intent, or necessitate domain-specific pretraining and heavyweight inference that disrupt service efficiency. More specifically, the domain-specific pretraining or fine-tuning concern mainly refers to language-to-item representation methods, where textual requests and items are aligned in a shared semantic space or interaction sequences are modeled in language space. In contrast, the heavyweight inference concern mainly refers to LLM-based reranking or reasoning methods; in this case, the additional cost comes from the request-aware inference procedure on top of the sequential backbone, rather than from the backbone itself. Our goal is not to replace periodic or continual retraining in production systems, but to provide short-horizon, request-conditioned behavioral adaptation between model updates, especially

---

†This work was completed during Zhiyuan's visit to Dalhousie University. [1]Gaoling School of Artificial Intelligence, Renmin University of China [2]Faculty of Computer Science, Dalhousie University. Correspondence to: Ga Wu <ga.wu@dal.ca>.

*Proceedings of the 43rd International Conference on Machine Learning*, Seoul, South Korea. PMLR 306, 2026. Copyright 2026 by the author(s).

when a user issues an immediate natural-language request or post-hoc feedback. This trade-off stems from a fundamental limitation: they treat the recommender backbone as a static black box. Crucially, they lack mechanisms for *posterior control*—the ability to intervene directly on the internal mechanics of a deployed model to enforce constraints not present during training. Unlike traditional user-controllable recommendation, which relies on ante-hoc conditioning, our setting demands modifying the behavior of a frozen sequential backbone on the fly. This requires moving beyond simple conditioning to actively steering the model's parameters in response to arbitrary and complex natural-language signals, a capability largely absent in current architectures.

We propose **C**ontrol via **R**equest-**A**ware **M**asking for **E**diting **R**ecommenders (**CRAMER**), a lightweight framework that treats a user's natural-language request as a control input and instantaneously modulates a frozen sequential recommender backbone via parameter masking. Drawing inspiration from model control theory (Li & Rush, 2020; Li et al., 2022), CRAMER applies learned masks to the backbone's parameters so the model's behavior is steered toward the requested intent with minimum computational overhead (Wen et al., 2016; Frankle & Carbin, 2019). CRAMER begins by mean-pooling across all token embeddings from the request to derive a faithful representation of the user's immediate request (Mosbach et al., 2020). A Gumbel–Top-$k$ step (Kool et al., 2019) then produces a sparse row–column gate vector, which is decomposed into per-matrix row and column gates and converted into entrywise masks applied to the selected matrices of the frozen Transformer-based sequential recommender. For robustness, CRAMER introduces three masking strategies for attention output matrices and feed-forward networks (FFNs) in the Transformer-based backbone. The training objective for the request-to-mask module combines a prediction loss with a KL regularizer that encourages the learned gate distribution to follow a sparsity prior (details in Appendix A). Empirically, this masking-based control achieves adaptation with minimal computational overhead, outperforms four state-of-the-art request-aware baselines on multiple large-scale benchmarks, offering a practical, scalable paradigm for request-aware sequential recommendation.

## 2. Background and Related Work

Sequential recommendation aims to predict the next interaction from historical sequences of consumed items, capturing the temporal dynamics beyond static profiles (Pan et al., 2024). In the past few years, Transformer-based models have become the predominant approach (Fang et al., 2020), among which SASRec (Kang & McAuley, 2018) and BERT4Rec (Sun et al., 2019) are the most representative, consistently serving as strong baselines across diverse

scenarios (Zivic et al., 2024).

Yet, current approaches struggle to adapt when users express immediate intent through natural-language requests (Li et al., 2023). In practice, an immediate request can emphasize aspects of prior preferences, or explicitly negate them (Wu et al., 2019; Luo et al., 2020), thereby calling for models that can adapt dynamically rather than relying solely on static long-term signals. Prior request-aware approaches fall into three strands: (i) request augmentation (He et al., 2022), which conditions sequential models on user-generated tags or requests to capture short-term intent but relies on shallow representations; (ii) language-to-item representations, which pretrain encoders to bridge natural language and items (Hou et al., 2024) or model sequences directly in language space (Li et al., 2023), improving coverage and transfer but requiring domain-specific pretraining or fine-tuning and potentially adding inference cost; and (iii) LLM-based methods, which leverage large language models for recommendation via semantic enhancement (Liu et al., 2024), constrained generation (Liao et al., 2026), listwise reasoning re-rankers (Zhang et al., 2025), or dynamic user-intent prompting (Xu et al., 2025), though effective, they are often overly complex and demand high computation and latency. These limitations motivate lightweight mechanisms that condition strong sequential backbones on immediate requests.

In the request-aware sequential recommendation mentioned above, retraining or fully fine-tuning complex Transformer-based backbones is computationally prohibitive for real-time adaptation. Since these models already encode long-term preference signals, it is common to keep the backbone frozen and introduce lightweight modules for parameter-efficient adaptation (Su et al., 2025; Shen et al., 2025), as in natural language processing (NLP) (Son et al., 2025) and computer vision (CV) (Qin et al., 2024). Such approaches include prefix/prompt tuning (Li & Liang, 2021; Lester et al., 2021), which prepends small vectors for only coarse control; adapter modules (Houlsby et al., 2019), which insert trainable layers but increase latency; and latent token insertion (Sun et al., 2025), which offers flexible conditioning at the cost of additional parameters. Masking methods instead stand out by learning task-dependent masks over weights or activations, enabling reversible and fine-grained control without retraining (Zhao et al., 2020; Ansell et al., 2022; Litschko et al., 2022; Tao et al., 2023; Svirsky et al., 2024), though their potential for conditioning on natural-language requests remains underexplored. Overall, prior work on parameter-efficient adaptation primarily targets task- or domain-level transfer, whereas CRAMER enables instance-level, request-conditioned posterior control of a frozen sequential recommender at inference time.

# 3. Methodology

## 3.1. Task Definition

Let $\mathcal{U}$ and $\mathcal{I}$ denote the sets of users and items, respectively. For a user $u \in \mathcal{U}$, we represent the historical interaction sequence as $\boldsymbol{s}_u = (i_1, \ldots, i_T)$ with $i_t \in \mathcal{I}$, $t \in \{1, 2, \ldots, T\}$, and denote the ground-truth next item by $i_{T+1}^\star \in \mathcal{I}$. In addition to these common notations of sequential recommendation, in the request-aware scenario, at time step $T+1$, the user $u$ provides a natural-language request $\mathbf{q}_u$, which specifies the user's immediate intent.

We consider a trained sequential recommender $f_\theta$ with parameters $\theta$. Given the interaction sequence $\boldsymbol{s}_u$ and request $\mathbf{q}_u$ of user $u$, the model scores each $i \in \mathcal{I}$ and predicts

$$\hat{i}_{T+1} \;=\; \arg\max_{i \in \mathcal{I}} \; f_\theta(i \mid \boldsymbol{s}_u, \mathbf{q}_u). \tag{1}$$

Ideally, we want this prediction to coincide with the ground-truth next item, i.e., $\hat{i}_{T+1} = i_{T+1}^\star$. The overall training objective is to maximize the total conditional log-likelihood of ground-truth next items over all users, i.e.,

$$\max \sum_{u \in \mathcal{U}} \log p\big(i_{T+1}^\star \mid \boldsymbol{s}_u, \mathbf{q}_u; \theta\big), \tag{2}$$

which amounts to encouraging the model to assign the highest probability to the ground-truth next item $i_{T+1}^\star$ given both the historical sequence and the accompanying request, consistent with the ideal case of Equation (1). In contrast to conventional sequential recommendation that relies solely on $\boldsymbol{s}_u$ and the backbone $f_\theta$, this formulation explicitly incorporates $\mathbf{q}_u$, allowing the model to reconcile long-term preferences with immediate intent.

To optimize Objective (2), existing methods take three main routes. Some manipulate $\boldsymbol{s}_u$, e.g., by augmenting or transforming it with $\mathbf{q}_u$ (He et al., 2022; Li et al., 2023; Hou et al., 2024; Liu et al., 2024), but such strategies often yield shallow control. Others introduce auxiliary request-aware modules that fuse with the backbone (Liao et al., 2026; Zhang et al., 2025), at the cost of added latency and complexity. A more direct option is to fine-tune or retrain the backbone parameters $\theta$ based on $\mathbf{q}_u$, but this is computationally expensive and impractical—since $\theta$ is often trained, deployed, and frozen in practice. Thus, the key challenge is to control the frozen sequential recommender backbone model given $\mathbf{q}_u$.

This motivates a mapping $\mathcal{F}_\phi$ with trainable parameters $\phi$, which transforms $\mathbf{q}_u$ into the control signal vector $\boldsymbol{m} \in \mathbb{R}^d$. Then, we apply $\boldsymbol{m}$ to $\theta$ through a series of operations $C_{\boldsymbol{m}}(\theta)$ to obtain the edited parameters $\theta'$. Therefore, starting from Objective (2), we can rewrite our goal as finding

$$\phi^\star = \arg\max_\phi \sum_{u \in \mathcal{U}} \log p\big(i_{T+1}^\star \mid \boldsymbol{s}_u, \theta'\big), \tag{3}$$

where $\theta' = C_{\boldsymbol{m}}(\theta)$, $\boldsymbol{m} = \mathcal{F}_\phi(\mathbf{q}_u)$.

## 3.2. Variational Motivation for Model Control

Equation (3) defines the target optimization goal for request-aware sequential recommendation. Exact marginalization over all control signals is intractable; therefore, we adopt a variational perspective (Blei et al., 2017) primarily as a conceptual guide for designing a sparse, request-conditioned controller, rather than as a strict inference procedure that CRAMER aims to optimize exactly. In particular, our practical algorithm does not perform full variational inference over control signals, but instead leverages this perspective to motivate the use of structured gating and sparsity, inducing regularization conditioned on natural-language requests.

**Variational Lower Bound.** Because the control signals in $\boldsymbol{m}$ are actually designed to be binary (in the form of gates, details in later sections), we adopt a factorized Bernoulli prior $p(\boldsymbol{m})$ and approximate the posterior with a mean-field Bernoulli distribution $Q_\phi(\boldsymbol{m} \mid \mathbf{q}_u)$ parameterized by $\phi$ (Equation (A.4)); see Appendix A for details. This Bernoulli parameterization is natural because the control signal is a binary row–column gate vector. We use a factorized Bernoulli prior because it matches the hard sparsity budget, admits a closed-form KL term, and keeps the regularization cost linear in the gate dimension. We use a mean-field Bernoulli posterior for the same tractability reason: the request embedding is mapped to gate logits, which define lightweight request-conditioned gate probabilities. Although the KL term assumes independent Bernoulli gates, the practical controller is not fully independent: all gate logits are generated jointly from the same request representation, and the final hard mask is further coupled by the exact $k$-hot Gumbel–Top-$k$ selection. Consider a single user $u \in \mathcal{U}$ in Equation (3), for such $Q_\phi(\boldsymbol{m} \mid \mathbf{q}_u)$ and $p(\boldsymbol{m})$, the marginal likelihood admits the variational lower bound:

$$\log p\big(i_{T+1}^\star \mid \boldsymbol{s}_u, \theta'\big) \;\geq\;$$
$$\int Q_\phi(\boldsymbol{m} \mid \mathbf{q}_u) \log p\big(i_{T+1}^\star \mid \boldsymbol{s}_u, \theta'\big) \, \mathrm{d}\boldsymbol{m} \quad (4)$$
$$- \, \mathrm{KL}[Q_\phi(\boldsymbol{m} \mid \mathbf{q}_u) \,\|\, p(\boldsymbol{m})],$$

with the evidence lower bound (ELBO)

$$\mathcal{L}_{\mathrm{ELBO}}(u) = \mathbb{E}_{\boldsymbol{m} \sim Q_\phi}\Big[ \log p\big(i_{T+1}^\star \mid \boldsymbol{s}_u, \theta'\big) \Big] \\ - \, \mathrm{KL}\Big(Q_\phi(\boldsymbol{m} \mid \mathbf{q}_u) \,\|\, p(\boldsymbol{m})\Big). \tag{5}$$

For the detailed derivation of Equation (5), please refer to Appendix A. In practice, we approximate it using Gumbel–Top-$k$ sampling with a straight-through estimator (see Sections 3.3 and 3.4), which is a more straightforward approach and better suited for recommender systems.

**Training Objective.** Motivated by Equation (5), we construct a tractable surrogate training objective with a KL

term that admits a closed-form expression (Equation (A.5)) under the factorized Bernoulli assumption. We denote by $\ell(\hat{i}_{T+1}, i^\star_{T+1})$ the predictive loss, i.e., the original training loss used by the backbone recommender. Our training objective $\mathcal{L}(\phi)$ is designed as

$$\frac{1}{|\mathcal{U}|} \sum_{u \in \mathcal{U}} \Big[ \underbrace{\ell(\hat{i}^{(u)}_{T+1}, i^{\star(u)}_{T+1})}_{\text{predictive loss}} + \lambda_{\text{KL}} \cdot \underbrace{\frac{1}{d} \text{KL}[\, Q \,\|\, p \,]}_{\text{KL regularizer}} \Big]. \quad (6)$$

where $d$ is the dimension of $\boldsymbol{m}$. The Objective (6) consists of two complementary terms. The first is the predictive loss, which directly drives the model to rank the ground-truth item highest given the historical sequence and request, ensuring recommendation accuracy. The second is the KL regularizer, which encourages the posterior distribution $Q_\phi$ to stay close to the sparsity prior $p(\boldsymbol{m})$, thereby enforcing compact and stable control. Note that we divide the KL term by the dimension $d$ of $\boldsymbol{m}$ to normalize its scale, preventing it from dominating as $d$ increases and ensuring a balanced trade-off between predictive accuracy and sparsity control. We emphasize that Objective (6) is a variationally inspired surrogate rather than a strict ELBO. The variational view motivates sparse request-conditioned control and the Bernoulli KL regularizer, while the practical forward controller uses an exact hard $k$-hot mask sampled by Gumbel–Top-$k$ with a straight-through estimator. Therefore, the KL term is computed on the Bernoulli relaxation of the gates and serves as an auxiliary sparsity-inducing regularizer aligned with the hard budget.

Based on Objective (6), we propose Control via Request-Aware Masking for Editing Recommenders (CRAMER). At a high level, CRAMER adapts a frozen sequential recommender to natural-language requests by learning lightweight binary gate vectors that map each request to masks over a pretrained backbone. The edited model fuses long-term preferences with immediate intent, enabling rapid, no-retraining adaptation while preserving fine-grained control. Figure 1 overviews the framework and the following sections detail its components.

### 3.3. Request-to-Mask Adaptation

As discussed in Section 2, masking parameters of a deep model provides a lightweight but expressive mechanism for model control. In CRAMER, masking should be viewed as a budgeted behavior-editing mechanism rather than merely as feature suppression. A sparse request-conditioned mask selectively activates or deactivates existing computation paths in the frozen sequential recommender backbone, thereby steering pre-learned preference representations toward the current request without relearning dense query-aware representations from scratch. Hard row–column masking is particularly aligned with our setting because it provides

an exact $k$-hot sparsity budget and keeps the training-time controller consistent with the low-overhead inference mechanism used at deployment. In this section, we introduce how CRAMER converts natural-language requests into masks that are used to control the backbone.

**Request Embedding.** We begin by describing how CRAMER encodes a natural-language request into an embedding that conditions the recommendation model (frozen sequential recommender backbone) $f_\theta$. Unlike historical interaction sequences, requests are diverse and may contain negations, constraints, or attribute-specific preferences. To extract their semantics, we introduce a lightweight request encoder $\text{E}_{\phi_{\text{enc}}}$ based on a pretrained language model (PLM). Given a request $\mathbf{q}_u$, we tokenize it and obtain contextualized token embeddings, which are mean-pooled across all tokens to form a stable representation (Mosbach et al., 2020). We adopt mean pooling because it is simple, stable, and architecture-agnostic for variable-length requests. Formally,

$$\boldsymbol{e}_q = \text{E}_{\phi_{\text{enc}}}(\mathbf{q}_u) \in \mathbb{R}^h,$$

where $h$ is the hidden dimension. The resulting semantic embedding $\boldsymbol{e}_q$ summarizes the user's immediate intent, allowing CRAMER to capture request semantics in a modular form while remaining compatible with the frozen sequential recommender backbone.

**Defining the Controllable Subset.** For a Transformer-based sequential recommender system $f_\theta$, we identify a subset of its parameters as *controllable* subset $\theta_M$ that is crucial and suitable for being masked. In Transformer architectures, FFNs constitute the majority of parameters and act as key–value memories (Geva et al., 2020; Gerber, 2025); selectively masking them directly modulates what the model "remembers." Moreover, attention heads often exhibit redundancy (Michel et al., 2019), and the multi-head attention (MHA) output projection matrices $W_O$ aggregate head outputs into the residual stream (Hu et al., 2022), so masking $W_O$ provides a compact, high-leverage control knob. Guided by these observations, CRAMER supports three scopes for $\theta_M$: (i) FFNs-only, (ii) $W_O$-only, and (iii) FFNs $+ W_O$. This choice is not meant to rule out other Transformer components, but to select natural intervention points that transform intermediate features and recombine attention-head outputs while still supporting structured row–column masking with low overhead. Formally, we write the maskable set of $L$ matrices as

$$\theta_M = \big\{ \boldsymbol{W}^{(l)} \in \mathbb{R}^{\alpha_l \times \beta_l} \big\}_{l=1}^L.$$

**Projection to Gate Logits.** Instead of assigning a mask to every parameter in $\theta_M$—which would incur prohibitive overhead given the scale of Transformer backbones—our scheme performs gating at the row and column levels (Svirsky et al., 2024). This structured design drastically

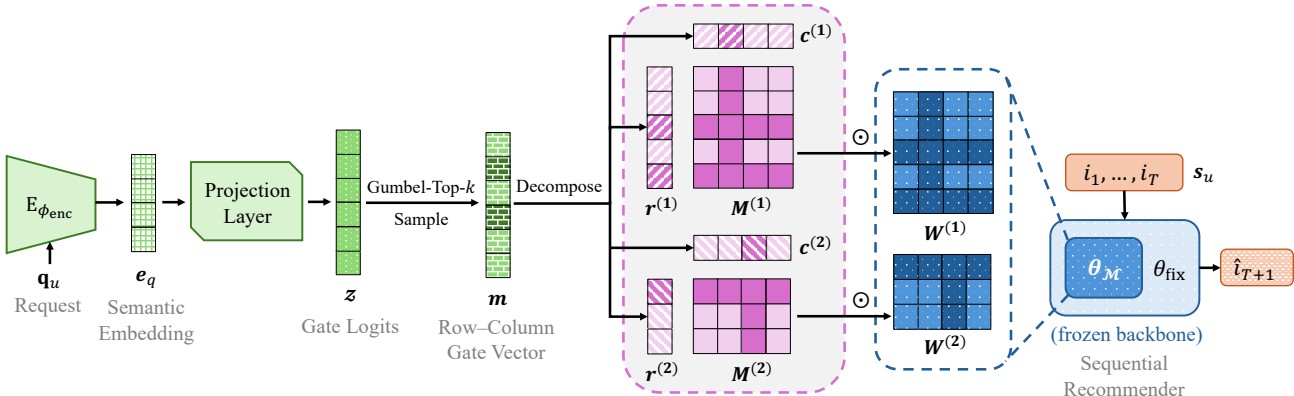

*Figure 1.* The overview of the proposed CRAMER framework. The figure shows the whole process of CRAMER converting a natural-language request into masks and controlling the sequential recommender (frozen sequential recommender backbone). The gray area with a pink dashed border represents the "Row–Column Gating Masks" paragraph in Section 3.3.

reduces the number of trainable parameters while providing fine-grained, lightweight control over the backbone. Given the semantic embedding $e_q$, we first map it to gate logits via a linear projection layer:

$$z = W_{\text{proj}} e_q + b_{\text{proj}} \in \mathbb{R}^d,$$

where $d = \sum_{l=1}^{L} (\alpha_l + \beta_l)$ is the total number of row and column dimensions under the chosen scope, and $(W_{\text{proj}}, b_{\text{proj}})$ are trainable parameters.

**Constructing Sparse Binary Vector.** To achieve lightweight yet effective control, we constrain the binary vector to be $k$-hot. Let $\rho \in (0, 1)$ be the drop ratio and retain exactly $k = \lceil (1-\rho)d \rceil$ active entries. To obtain them, we employ the Gumbel–Top-$k$ trick (Kool et al., 2019): for each coordinate $i$, we sample $g_i \sim \text{Gumbel}(0)$ and form

$$\tilde{z}_i = z_i + g_i, \qquad i = 1, \dots, d.$$

The indices of the $k$ largest $\tilde{z}_i$ form $S_k$, and the activated entries are

$$m_i = \mathbb{I}\{i \in S_k\}, \qquad m \in \{0,1\}^d. \quad (7)$$

This binary vector $m$ is precisely the instantiation of the control signal vector mentioned in Equation (3), and the first four paragraphs of this section together constitute a concrete realization of the mapping $\mathcal{F}_\phi$ described in Section 3.1.

**Row–Column Gating Masks.** In our CRAMER framework, $m$ acts as a row-column gate vector that compactly specifies the activations of all maskable matrices in $\theta_M$. We decompose $m$ into per-matrix segments to obtain, for each $l$, a row gate vector $r^{(l)} \in \{0,1\}^{\alpha_l}$ and a column gate vector $c^{(l)} \in \{0,1\}^{\beta_l}$, and define the entrywise mask

$$M_{ij}^{(l)} = r_i^{(l)} \cdot c_j^{(l)}, \quad 1 \le i \le \alpha_l, \, 1 \le j \le \beta_l.$$

Collecting all $M^{(l)}$ and applying them entrywise to the corresponding $W^{(l)} \in \theta_M$ yields the edited backbone $f_{\theta'}$, with parameters

$$\theta' = (\theta_{/M}, \{W^{(l)} \odot M^{(l)} : W^{(l)} \in \theta_M\}). \quad (8)$$

where $\theta_{/M}$ represents the parameters in $\theta$ except $\theta_M$. The operations in this paragraph are the specific form of $C_m(\theta)$ mentioned in Equation (3). Thus, the semantic embedding $e_q$ is converted into a structured set of row-column gating masks that modulate the frozen sequential recommender backbone with minimal overhead while retaining fine-grained control.

### 3.4. Learnable Components and Discrete Optimization

**Trainable Parameters.** Since the backbone $f_\theta$ is frozen, training updates only the request-to-mask (Section 3.3) module. Two components are learnable: (i) the projection layer $(W_{\text{proj}}, b_{\text{proj}})$ that maps the semantic embedding $e_q$ to gate logits $z$, and (ii) a subset $\phi_t$ of the request encoder parameters $\phi_{\text{enc}}$ (initialized from a PLM). We consider three regimes for $\phi_t \subseteq \phi_{\text{enc}}$: none (encoder fully frozen), last (fine-tune the last layer only), and all (end-to-end tuning). This offers a flexible way to balance adaptation capacity and efficiency across backbones and datasets.

**Straight-Through Training for Gating.** As we obtain $m$ by discretely sampling $z$ (see Equation (7)), this process blocks gradients from $m$ to logits $z$. To address this issue, we adopt a straight-through estimator (STE) (Bengio et al., 2013; Jang et al., 2016) with a temperature-controlled soft surrogate in the backward pass. Concretely, the forward pass uses hard $k$-hot gates from Gumbel–Top-$k$, while the backward pass propagates gradients through

$$v_i = \frac{\exp(z_i/\tau)}{\sum_{j=1}^{d} \exp(z_j/\tau)}, \qquad i = 1, \dots, d, \; \tau > 0,$$

serving as a continuous relaxation of the binary $m_i$. This STE scheme enables end-to-end optimization of the trainable parameters $(\boldsymbol{W}_{\text{proj}}, \boldsymbol{b}_{\text{proj}}, \phi_t)$ despite the discrete sampling of $\boldsymbol{m}$.

# 4. Experiments and Evaluation

The implementation of CRAMER is currently available at `https://github.com/zhiyuansu0326/CRAMER-ICML2026`.

## 4.1. Experimental Setup

**Datasets.** We consider four representative datasets: (i) **Re-Dial** (Li et al., 2018), a conversational recommendation dataset with about 11.3K movie-recommendation dialogues, where users explicitly mention movies and annotate whether they have seen or liked them; (ii) **KuaiSAR** (Sun et al., 2023), a large-scale short-video interaction dataset from Kuaishou that captures both search and recommendation behaviors, containing about 19.6M actions and 6.9M items; (iii) **Beauty**, a subset of the Amazon Reviews (Hou et al., 2024) data, with approximately 701.5K reviews and 112.6K items, enriched in metadata, review text and timestamps; (iv) **CDs&Vinyl**, another subset from Amazon Reviews (Hou et al., 2024), containing about 4.8M reviews and 701.7K items, also possessing metadata, review text and timestamps. We preprocess the four datasets in a unified manner. Limited by the computing budget, we downsample the three larger datasets (KuaiSAR, Beauty, CDs&Vinyl). For more information on data preprocessing and statistics, see the Appendix B.1.

**Backbones.** We adopt two widely used Transformer-based sequential recommenders as frozen sequential recommender backbones. (i) **SASRec** (Kang & McAuley, 2018) employs unidirectional self-attention to model sequential dependencies in user interaction histories with high efficiency, and has become a standard baseline in sequential recommendation. (ii) **BERT4Rec** (Sun et al., 2019) adopts a bidirectional Transformer trained with a masked item prediction objective, enabling the model to capture both left and right contexts of a target position and to produce context-rich sequence representations. These two models represent the most established architectures for sequential recommendation and provide strong non–request-aware references for our study.

**Baselines.** On top of the frozen sequential recommender backbones, we further compare CRAMER with several state-of-the-art request-aware methods that incorporate natural-language requests. (i) **Query-SeqRec** (He et al., 2022) introduces a request encoder to represent the query and injects it into the backbone's sequential representation through concatenation or attention, enabling request-conditioned rel-

evance scoring. (ii) **BLaIR** (Hou et al., 2024) encodes both request text and item metadata into a shared semantic space to compute similarity signals, which are then fused with the backbone's outputs to enhance ranking with request-aware semantics. (iii) **LLM-ESR** (Liu et al., 2024) leverages cached LLM-derived semantic embeddings and combines them with collaborative backbone embeddings via a lightweight adapter, providing notable benefits for long-tail users and items while keeping the backbone frozen. (iv) **REARANK** (Zhang et al., 2025) first generates an initial ranking using the backbone and then applies an LLM reranker that reasons over user history, the request, and candidate metadata to refine the list, combining sequential modeling with listwise reasoning. The integration details of these baselines with the frozen sequential recommender backbones are given in Appendix B.9.

**Optional PLMs.** As mentioned in Section 3.3, the request encoder $\text{E}_{\phi_{\text{enc}}}$ is initialized from a PLM based on Transformer. We consider four PLMs to cover the efficiency–accuracy spectrum: (i) **BERT style** (Devlin et al., 2019), a classic encoder. (ii) **RoBERTa style** (Liu et al., 2019), a robust medium encoder. (iii) **MiniLM style** (Wang et al., 2020), a very lightweight encoder. (iv) **ModernBERT style** (Warner et al., 2024), a modern high-capacity base encoder. All encoders use default text preprocessing, and we apply mean-pooling over the final hidden states (Section 3.3) to produce the semantic embedding $\boldsymbol{e}_q$, which we find more stable than single-token pooling when handling diverse request phrasing (Mosbach et al., 2020).

**Evaluation Metrics.** We adopt widely used ranking metrics in recommender system evaluation. Specifically, we report **HR@$k$** (Hit Ratio), **NDCG@$k$** (Normalized Discounted Cumulative Gain) and **MRR** (Mean Reciprocal Rank) at cutoff values $k \in \{10, 20\}$. These are very commonly used metrics in recommender system evaluation.

## 4.2. Overall Performance

Following previous papers (Kang & McAuley, 2018; Liu et al., 2024), we randomly sample 100 items that the user has not interacted with as the negatives paired with the only ground-truth positive for calculation of the metrics. Table 1 reports the overall performance of four baselines and our proposed CRAMER on four benchmark datasets under two frozen Transformer backbones.

**Aggregate Comparison.** From the results, CRAMER consistently outperforms all baselines across datasets and metrics under both SASRec and BERT4Rec backbones. Intuitively, we observe that CRAMER achieves the best results on all metrics across all experiments. After applying Benjamini–Hochberg (BH) procedure (FDR = 0.05) across all 48 paired t-tests, CRAMER remains significantly better than the strongest baseline in 41 settings (85.42%), indicat-

*Table 1.* Overall results of four baselines and our CRAMER. H@$k$, N@$k$, and M@$k$ denote HR@$k$, NDCG@$k$, and MRR@$k$, respectively (averaged over five runs). For each setting, the boldface refers to the highest result, and the underline indicates the second best result. "*" marks statistically significant improvements after BH procedure (FDR = 0.05) across all 48 t-tests.

| Method | ReDial | | | | | | KuaiSAR | | | | | |
|---|---|---|---|---|---|---|---|---|---|---|---|---|
| | H@10 | H@20 | N@10 | N@20 | M@10 | M@20 | H@10 | H@20 | N@10 | N@20 | M@10 | M@20 |
| **SASRec** | 0.426 | 0.573 | 0.373 | 0.410 | 0.344 | 0.354 | 0.430 | 0.601 | 0.346 | 0.389 | 0.306 | 0.318 |
| +Query-SeqRec | 0.450 | 0.596 | 0.391 | 0.429 | 0.350 | 0.361 | 0.451 | 0.567 | 0.348 | 0.378 | 0.313 | 0.322 |
| +BLaIR | 0.447 | 0.582 | 0.392 | 0.426 | 0.287 | 0.296 | 0.479 | 0.612 | 0.408 | 0.443 | 0.293 | 0.303 |
| +LLM-ESR | 0.516 | 0.666 | 0.385 | 0.423 | 0.323 | 0.334 | 0.496 | 0.628 | 0.392 | 0.427 | 0.343 | 0.351 |
| +REARANK | 0.549 | 0.684 | 0.414 | 0.449 | 0.408 | 0.417 | 0.538 | 0.645 | 0.409 | 0.437 | 0.366 | 0.374 |
| +CRAMER (Ours) | **0.578*** | **0.694*** | **0.428*** | **0.456*** | **0.413** | **0.421** | **0.556*** | **0.748*** | **0.436*** | **0.484*** | **0.391*** | **0.405*** |
| **BERT4Rec** | 0.421 | 0.542 | 0.355 | 0.387 | 0.272 | 0.281 | 0.436 | 0.591 | 0.366 | 0.407 | 0.311 | 0.322 |
| +Query-SeqRec | 0.462 | 0.563 | 0.347 | 0.373 | 0.307 | 0.315 | 0.464 | 0.577 | 0.364 | 0.393 | 0.349 | 0.358 |
| +BLaIR | 0.466 | 0.654 | 0.395 | 0.442 | 0.333 | 0.348 | 0.480 | 0.652 | 0.401 | 0.445 | 0.339 | 0.352 |
| +LLM-ESR | 0.515 | 0.668 | 0.427 | 0.465 | 0.358 | 0.369 | 0.530 | 0.715 | 0.389 | 0.436 | 0.324 | 0.338 |
| +REARANK | 0.536 | 0.680 | 0.388 | 0.424 | 0.355 | 0.366 | 0.566 | 0.691 | 0.416 | 0.448 | 0.355 | 0.364 |
| +CRAMER (Ours) | **0.580*** | **0.753*** | **0.451*** | **0.497*** | **0.376*** | **0.389*** | **0.598*** | **0.717** | **0.434*** | **0.467*** | **0.382*** | **0.390*** |

| Method | Beauty | | | | | | CDs&Vinyl | | | | | |
|---|---|---|---|---|---|---|---|---|---|---|---|---|
| | H@10 | H@20 | N@10 | N@20 | M@10 | M@20 | H@10 | H@20 | N@10 | N@20 | M@10 | M@20 |
| **SASRec** | 0.442 | 0.574 | 0.385 | 0.419 | 0.338 | 0.348 | 0.480 | 0.658 | 0.398 | 0.444 | 0.360 | 0.374 |
| +Query-SeqRec | 0.479 | 0.606 | 0.352 | 0.384 | 0.302 | 0.313 | 0.511 | 0.698 | 0.406 | 0.454 | 0.355 | 0.370 |
| +BLaIR | 0.495 | 0.622 | 0.421 | 0.453 | 0.348 | 0.357 | 0.525 | 0.627 | 0.450 | 0.477 | 0.349 | 0.357 |
| +LLM-ESR | 0.503 | 0.701 | 0.445 | 0.495 | 0.379 | 0.395 | 0.560 | 0.699 | 0.434 | 0.470 | 0.346 | 0.355 |
| +REARANK | 0.548 | 0.681 | 0.474 | 0.509 | 0.335 | 0.344 | 0.612 | 0.719 | 0.454 | 0.481 | 0.378 | 0.385 |
| +CRAMER (Ours) | **0.574*** | **0.735*** | **0.489*** | **0.531*** | **0.385** | **0.397** | **0.619** | **0.726*** | **0.472*** | **0.498*** | **0.397*** | **0.404*** |
| **BERT4Rec** | 0.409 | 0.551 | 0.331 | 0.368 | 0.323 | 0.334 | 0.416 | 0.547 | 0.300 | 0.334 | 0.291 | 0.301 |
| +Query-SeqRec | 0.434 | 0.534 | 0.357 | 0.382 | 0.334 | 0.341 | 0.459 | 0.614 | 0.330 | 0.371 | 0.283 | 0.292 |
| +BLaIR | 0.459 | 0.627 | 0.364 | 0.407 | 0.315 | 0.327 | 0.462 | 0.617 | 0.412 | 0.452 | 0.333 | 0.345 |
| +LLM-ESR | 0.493 | 0.594 | 0.346 | 0.373 | 0.319 | 0.327 | 0.503 | 0.692 | 0.438 | 0.487 | 0.311 | 0.325 |
| +REARANK | 0.509 | 0.693 | 0.379 | 0.426 | 0.331 | 0.346 | 0.571 | 0.684 | 0.423 | 0.452 | 0.367 | 0.376 |
| +CRAMER (Ours) | **0.539*** | **0.734*** | **0.396*** | **0.447*** | **0.345*** | **0.359*** | **0.583*** | **0.695** | **0.487*** | **0.516*** | **0.433*** | **0.441*** |

ing consistent and robust improvements. This demonstrates that the request-aware masking mechanism effectively augments sequential recommenders, delivering more accurate predictions without requiring full fine-tuning.

**Results by Dataset and Backbone.** Across datasets, CRAMER shows clear advantages, particularly on large-scale datasets like KuaiSAR and CDs&Vinyl, where it substantially outperforms embedding-based and generative baselines, showing its ability to integrate long-term preferences with immediate requests. On smaller datasets (e.g., ReDial), it still yields consistent gains, underscoring robustness. Across backbones, CRAMER delivers stable improvements on both SASRec and BERT4Rec by incorporating request semantics, thereby addressing the limitation that both models rely solely on users' historical interactions for prediction. Overall, CRAMER is a general and effective framework for request-aware sequential recommendation across datasets and architectures.

**Intuitive User Case Study.** To provide a more intuitive understanding of how our approach improves recommendation quality, we further conduct a case study on five specific users from the CDs&Vinyl dataset. For more details, please refer to Appendix B.8.

**Summary.** In summary, CRAMER consistently outperforms embedding-based, generative, and reasoning-driven request-aware baselines. It achieves significant improvements across nearly all datasets, metrics, and backbones. The results in Table 1 establish CRAMER as a general, flexible and robust framework that effectively integrates long-term preferences with immediate intent while remaining efficient under frozen sequential backbones.

## 4.3. Sensitivity Analysis of Hyperparameters

While the overall results in Section 4.2 demonstrate the effectiveness of CRAMER, it is important to understand how different hyperparameters influence performance. We therefore conduct several sensitivity studies to disentangle the contribution of each design choice. In each backbone

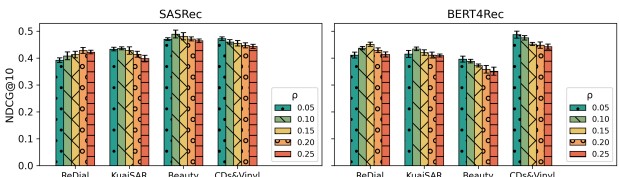

*Figure 2.* Sensitivity of CRAMER to drop ratio $\rho$, evaluated using NDCG@10. For each setting, five evaluations were performed, the column height shows its average result.

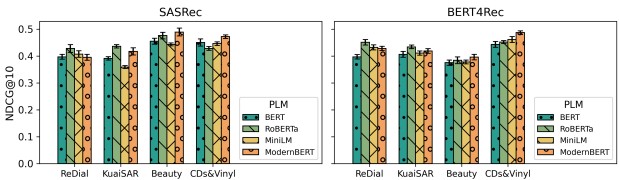

*Figure 3.* Impact of different PLMs on request encoding, evaluated using NDCG@10. For each setting, five evaluations were performed, the column height shows its average result.

$\times$ dataset experiment, we vary only the hyperparameter of interest while keeping all other settings fixed at their optimal values (listed in Appendix B.3). Evaluation is reported in terms of NDCG@10. Among the full set of hyperparameters we examined, two are particularly crucial: (i) drop ratio $\rho$, which determines how many units remain active under the request-to-mask mechanism; and (ii) selection of PLM used to initialize the request encoder. In this section we focus on these two factors, while additional experiments (e.g., experiments on $\theta_M$, $\lambda_{\mathrm{KL}}$, $\phi_t$) are deferred to Appendix B.4.

**Sensitivity to Drop Ratio $\rho$.** Figure 2 reports results for $\rho \in \{0.05, 0.10, 0.15, 0.20, 0.25\}$. We find that performance is generally robust within a moderate range but deteriorates at extreme values. In most cases, a $\rho$ of around 0.10 achieves the best balance: too small a $\rho$ (i.e., almost no parameters are masked) weakens the influence of request conditioning, while too large a $\rho$ (i.e., masking too many parameters) harms the backbone's capacity. We also find that small-scale datasets tend to achieve their best performance at larger values of $\rho$, while large-scale, information-dense datasets show optimal performance at smaller $\rho$. From a theoretical analysis, we believe this is because the risk of overfitting is greater on small-scale datasets; higher sparsity imposes stronger regularization, helping to avoid overfitting and making the model focus only on the most salient request signals. On large-scale datasets, there is enough data to support more active parameters, so denser masks allow more backbone capacity to be utilized, improving expressiveness (Hoefler et al., 2021).

**Selection of PLM.** Figure 3 compares MiniLM, BERT, RoBERTa, and ModernBERT as optional PLMs, which are used to initialize the request encoders. Overall, RoBERTa and ModernBERT consistently yield the best performance:

*Table 2.* Inference efficiency comparison for SASRec backbone. Runtime and GPU memory usage are measured as average per-request cost under identical settings. "Vanilla Backbone" reports the backbone-only cost, and the remaining rows show the additional overhead introduced by each request-aware method.

| Method | Runtime (s) | GPU Memory (MiB) |
|---|---|---|
| **SASRec** | 0.033 | 2024.1 |
| +Query-SeqRec | +0.021 | +1587.5 |
| +BLaIR | +0.029 | +1125.4 |
| +LLM-ESR | +0.016 | +1236.2 |
| +REARANK | +9.256 | +9824.7 |
| +CRAMER (Ours) | +0.018 | +1355.6 |

RoBERTa excels on small-scale or linguistically diverse datasets such as ReDial and Beauty, while ModernBERT dominates on large-scale or information-dense datasets such as CDs&Vinyl and KuaiSAR. In contrast, MiniLM, though computationally efficient, underperforms due to its limited capacity, and vanilla BERT trails behind its stronger successors in most cases. This demonstrates the ability of the CRAMER framework to fully leverage better and more robust language models, marking its excellence in capturing the request information.

### 4.4. Efficiency and Overhead

Beyond effectiveness, CRAMER also exhibits lightweight inference behavior. Once the request-to-mask module is trained, processing a new request requires only a single forward pass with a lightweight projection and masking operation, resulting in minimal per-request overhead and making the method well suited for real-time recommendation. Table 2 reports the average inference cost measured by wall-clock runtime and peak GPU memory usage under identical settings. Compared to vanilla SASRec, CRAMER introduces only a 0.018s runtime overhead, ranking among the most efficient request-aware methods. Its inference cost is comparable to LLM-ESR and lower than Query-SeqRec and BLaIR, while maintaining one of the smallest GPU memory footprints. In contrast, REARANK incurs substantial overhead (over two orders of magnitude higher runtime) due to its listwise reasoning across multiple candidates. Overall, CRAMER achieves a favorable trade-off between accuracy and efficiency: once trained, it consistently improves recommendation quality while keeping inference time and memory consumption low, enabling practical and scalable real-time deployment. Inference efficiency results for BERT4Rec are provided in Appendix B.5.

### 4.5. Mask Interpretability

To evaluate whether CRAMER's request-conditioned masks reflect meaningful semantics, we conduct an interpretability study on the ReDial dataset. Using genre labels obtained

*Table 3.* The proportion of romance-related movies in the top-10 recommendations under different request types (100 users). The first row reports the baseline results without any request.

| Type | Request Text | Avg | Var |
|------|--------------|-----|-----|
| \ | No request | 0.286 | 0.0218 |
| (1) | *"I'd like a romantic comedy."* | 0.432 ↑ | 0.0244 |
| (2) | *"Something sweet and heartwarming."* | 0.345 ↑ | 0.0253 |
| (3) | *"I want an offbeat, slow-burn emotional drama."* | 0.312 ↑ | 0.0371 |
| (4) | *"Please avoid romantic movies."* | 0.135 ↓ | 0.0187 |
| (5) | *"Maybe something less focused on love."* | 0.204 ↓ | 0.0198 |
| (6) | *"Skip movies with amour-driven plots."* | 0.253 ↓ | 0.0389 |

via the Open Movie Database[1] (one of ReDial's original data sources), we determine whether a recommended movie belongs to the romance-related category. For a sampled group of 100 users, we issue six types of requests around the "romance" concept: (1) clear positive, (2) ambiguous positive, (3) rare-term positive, (4) clear opposite, (5) ambiguous opposite, and (6) rare-term opposite, to influence their recommendation results. For each request, we measure the proportion of romance-related items in top-10 list and compute the mean and variance across users.

Across all six request types, the results in Table 3 demonstrate that CRAMER produces consistent and semantically aligned shifts in the recommendation distribution. Clear positive requests substantially increase the proportion of romance-related movies in the top-10 list, while clear opposite requests consistently decrease it. This directional behavior provides evidence that CRAMER's request-conditioned masks can induce semantic shifts aligned with the intended request. Ambiguous requests also induce smaller but still noticeable shifts in the expected direction, indicating that CRAMER can interpret indirect user intent. Moreover, rare-term requests can still lead to appropriate adjustments. Although we can observe an increase in variance, it remains within acceptable limits, suggesting robustness to infrequent phrasing. These findings provide evidence that CRAMER can induce interpretable and stable request-conditioned shifts over the backbone model.

## 5. Conclusion

In this paper, we introduced CRAMER, a lightweight framework for request-aware sequential recommendation that treats natural-language requests as control inputs. We first formalized the problem and proposed a variationally motivated training objective with KL regularization and STE, enabling stable and efficient optimization. CRAMER encodes each request into a semantic embedding, which is projected into structured row–column masks that modulate frozen Transformer backbones, providing fine-grained and efficient control without retraining. Through extensive experiments on four benchmark datasets and two backbones,

we demonstrated that CRAMER consistently outperforms strong request-aware baselines across multiple metrics while incurring minimal runtime and memory overhead. Overall, CRAMER establishes a new paradigm for controllable, efficient, and scalable integration of immediate user intent into sequential recommendation.

## Impact Statement

This paper presents work whose goal is to advance the field of machine learning, with a focus on enabling efficient and flexible control of sequential recommender systems through natural-language requests. The proposed approach allows deployed recommender models to adapt their behavior to users' immediate intents without retraining, which may improve user experience and interaction quality in real-world recommendation services. In addition, by avoiding costly backbone fine-tuning or large language model inference at deployment time, this work has the potential to reduce computational overhead and energy consumption in large-scale systems.

At the same time, increased controllability of recommender systems may raise practical concerns related to unintended or inappropriate steering of recommendations, such as conflicts between short-term requests and users' long-term interests. In this work, the proposed method is designed to provide moderate, request-conditioned adjustments rather than complete overrides of learned preferences. We believe that responsible deployment of such techniques should be accompanied by appropriate system-level safeguards and human-centered design choices. Overall, the broader impacts of this work are consistent with those commonly encountered when advancing controllable and efficient machine learning systems.

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

## A. Derivation of the Training Objective

We start from the request-aware maximum-likelihood formulation in Objective (2):

$$\max \sum_{u \in \mathcal{U}} \log p\big(i_{T+1}^\star \mid \boldsymbol{s}_u, \theta'\big), \quad \text{where } \theta' = C_{\boldsymbol{m}}(\theta), \;\; \boldsymbol{m} = \mathcal{F}_\phi(\mathbf{q}_u).$$

According to our definition in Section 3.1, $f_\theta$ is a frozen sequential backbone with parameters $\theta$, and only the subset $\theta_M$ is subject to request-conditioned masking. Let $\boldsymbol{m} \in \{0,1\}^d$ denote the row-column gate vector (Section 3.3) that selects row/column activations for all matrices in $\theta_M$. In Section 3.2 we adopt variational inference as a motivation for a tractable surrogate. Below we present the ELBO derivation, and then clarify how our practical training aligns with it under hard $k$-hot control. In the original Inequality (4), we use integral to characterize the abstract "control signals", but $\boldsymbol{m} \in \{0,1\}^d$ is actually a binary vector, so in the derivation in this section, we use summation instead of integral.

**Marginal Likelihood as a Sum over Masks.** For a single user $u \in \mathcal{U}$, the conditional likelihood is marginalized over all gates in $\boldsymbol{m}$:

$$\begin{aligned}
\log p\big(i_{T+1}^\star \mid \boldsymbol{s}_u, \theta'\big) &= \log p\big(i_{T+1}^\star \mid \boldsymbol{s}_u, \mathbf{q}_u, \boldsymbol{m}; \theta\big) \\
&= \log \sum_{\boldsymbol{m} \in \{0,1\}^d} p\big(i_{T+1}^\star \mid \boldsymbol{s}_u, \mathbf{q}_u, \boldsymbol{m}; \theta\big) \; p(\boldsymbol{m}).
\end{aligned} \quad \text{(A.1)}$$

**Prior Distribution.** We use a request-agnostic sparsity prior. Aligned with Section 3.3, we consider a factorized Bernoulli prior with activation rate equal to the actual keep ratio:

$$p(\boldsymbol{m}) \;=\; \prod_{i=1}^{d} \pi_0^{m_i} (1 - \pi_0)^{1 - m_i}, \qquad \pi_0 := \frac{k}{d}, \quad \text{(A.2)}$$

so that the prior mean exactly matches the realized budget of $k = \lceil (1 - \rho)d \rceil$ active gates (note that $\pi_0$ may differ slightly from $1 - \rho$ due to the ceiling operation).

**Variational Lower Bound.** Introduce a request-conditioned variational distribution $Q_\phi(\boldsymbol{m} \mid \mathbf{q}_u)$, parameterized by $\phi$. Multiply and divide inside the sum by $Q_\phi$, and apply Jensen's inequality:

$$\begin{aligned}
\log p\big(i_{T+1}^\star \mid \boldsymbol{s}_u, \theta'\big) &= \log \sum_{\boldsymbol{m} \in \{0,1\}} Q_\phi(\boldsymbol{m} \mid \mathbf{q}_u) \frac{p\big(i_{T+1}^\star \mid \boldsymbol{s}_u, \mathbf{q}_u, \boldsymbol{m}; \theta\big) \; p(\boldsymbol{m})}{Q_\phi(\boldsymbol{m} \mid \mathbf{q}_u)} \\
&= \log \mathbb{E}_{\boldsymbol{m} \sim Q_\phi(\cdot \mid \mathbf{q}_u)} \left[ \frac{p\big(i_{T+1}^\star \mid \boldsymbol{s}_u, \theta'\big) \; p(\boldsymbol{m})}{Q_\phi(\boldsymbol{m} \mid \mathbf{q}_u)} \right] \\
&\geq \mathbb{E}_{\boldsymbol{m} \sim Q_\phi(\cdot \mid \mathbf{q}_u)} \big[\log p\big(i_{T+1}^\star \mid \boldsymbol{s}_u, \theta'\big) + \log p(\boldsymbol{m}) - \log Q_\phi(\boldsymbol{m} \mid \mathbf{q}_u)\big] \\
&= \underbrace{\mathbb{E}_{\boldsymbol{m} \sim Q_\phi}\Big[\log p\big(i_{T+1}^\star \mid \boldsymbol{s}_u, \theta'\big)\Big]}_{\text{prediction term}} - \underbrace{\mathbb{E}_{\boldsymbol{m} \sim Q_\phi}\Big[\log \frac{Q_\phi(\boldsymbol{m} \mid \mathbf{q}_u)}{p(\boldsymbol{m})}\Big]}_{\text{KL}[Q_\phi(\boldsymbol{m} \mid \mathbf{q}_u) \,\|\, p(\boldsymbol{m})]}.
\end{aligned}$$

The above is the complete derivation of Equation (5). Note that $p\big(i_{T+1}^\star \mid \boldsymbol{s}_u, \mathbf{q}_u, \boldsymbol{m}; \theta\big) = p\big(i_{T+1}^\star \mid \boldsymbol{s}_u, \theta'\big)$. Hence, the per-user evidence lower bound (ELBO) is

$$\mathcal{L}_{\text{ELBO}}(u) = \mathbb{E}_{\boldsymbol{m} \sim Q_\phi}\Big[\log p\big(i_{T+1}^\star \mid \boldsymbol{s}_u, \theta'\big)\Big] \; - \; \text{KL}[Q_\phi(\boldsymbol{m} \mid \mathbf{q}_u) \,\|\, p(\boldsymbol{m})]. \quad \text{(A.3)}$$

which is consistent with Equation (5). In our actual algorithm the forward controller is hard $k$-hot (Gumbel–Top-$k$) with STE for gradients (Section 3.4); therefore we treat the ELBO as motivation and use a variationally inspired surrogate objective (detailed below) rather than maximizing Equation (A.3) strictly.

**Parametrization and Analytic KL.** The request-to-mask module (Section 3.3) produces logits $\boldsymbol{z} = \boldsymbol{W}_{\text{proj}} \boldsymbol{e}_q + \boldsymbol{b}_{\text{proj}} \in \mathbb{R}^d$ from the semantic embedding $\boldsymbol{e}_q$. We adopt a mean-field Bernoulli parameterization for regularization and monitoring:

$$Q_\phi(\boldsymbol{m} \mid \mathbf{q}_u) = \prod_{i=1}^{d} \pi_i^{m_i} (1 - \pi_i)^{1 - m_i}, \qquad \pi_i = \sigma(z_i), \;\; \sigma(x) = \frac{1}{1 + e^{-x}}. \quad \text{(A.4)}$$

With the Bernoulli prior in Equation (A.2), the KL admits a closed form:

$$\mathrm{KL}[Q_\phi \,\|\, p] = \sum_{i=1}^{d} \left[ \pi_i \log \frac{\pi_i}{\pi_0} + (1 - \pi_i) \log \frac{1 - \pi_i}{1 - \pi_0} \right], \tag{A.5}$$

and we use the normalized version $\overline{\mathrm{KL}} = \frac{1}{d}\,\mathrm{KL}[Q_\phi\|\,p]$ so that the penalty does not scale with $d$.

At inference, $\boldsymbol{m}$ is instantiated as an exactly $k$-hot vector by (deterministic) Top-$k$ on $\boldsymbol{z}$ (we drop Gumbel noise). At training time, the forward path also uses hard $k$-hot masks via Gumbel–Top-$k$, while the backward path propagates gradients through a softmax surrogate with temperature (STE; Section 3.4). In parallel, we compute the Bernoulli parameters $\pi_i = \sigma(z_i)$ and apply the analytic Bernoulli–Bernoulli KL in Equation (A.5) as an auxiliary sparsity prior. We set the prior mean to the realized keep ratio, $\pi_0 = k/d$, aligning the prior with the hard budget used in the forward path. Because the forward sampler is $k$-hot while the KL assumes independent Bernoulli gates, the overall objective is not a strict ELBO; it is a variationally inspired surrogate consistent with our hard-sparsity design.

**From Likelihood to Supervised Loss.** We train with a supervised next-item loss $\ell(\cdot)$ in place of $-\log p(\cdot)$ (consistent with Section 3.2). Let $\theta' = \left(\theta_{/M}, \{\boldsymbol{W}^{(l)} \odot \boldsymbol{M}^{(l)} : \boldsymbol{W}^{(l)} \in \theta_M\}\right)$ denote the edited backbone obtained by applying the row–column masks (Section 3.3). A per-user surrogate objective is

$$\mathcal{L}(u; \phi) = \underbrace{\ell\big(f_{\theta'}(\boldsymbol{s}_u, \mathbf{q}_u), \, i_{T+1}^{\star(u)}\big)}_{\text{predictive loss under hard } k\text{-hot forward}} + \lambda_{\mathrm{KL}} \cdot \overline{\mathrm{KL}}(\boldsymbol{z}; \pi_0), \tag{A.6}$$

where $\overline{\mathrm{KL}}(\boldsymbol{z}; \pi_0)$ is computed from $\pi = \sigma(\boldsymbol{z})$ via Equation (A.5), and gradients through the discrete selection are enabled by STE with a temperature-controlled soft surrogate (Section 3.4). Equivalently, we can view Equation (A.6) as a single-sample Monte Carlo estimator of the predictive term (with the sample drawn by Gumbel–Top-$k$) plus an analytic KL regularizer computed on the logits.

**Final Training Objective.** Aggregating and averaging Equation (A.6) over $u \in \mathcal{U}$ yields the training objective reported in Section 3.2:

$$\mathcal{L}(\phi) = \frac{1}{|\mathcal{U}|} \sum_{u \in \mathcal{U}} \left[ \underbrace{\ell(\hat{i}_{T+1}^{(u)}, i_{T+1}^{\star(u)})}_{\text{predictive loss}} + \lambda_{\mathrm{KL}} \cdot \underbrace{\overline{\mathrm{KL}}(\boldsymbol{z}; \pi_0)}_{\substack{\text{KL regularizer} \\ \text{with prior mean } \pi_0}} \right].$$

The first term provides supervision for request-aware prediction under the edited backbone $f_{\theta'}$, while the second acts as an auxiliary sparsity prior that stabilizes optimization and prevents degenerate dense masks. In summary, the variational view motivates a tractable, analytically regularized surrogate that preserves strict $k$-sparsity in the forward path (via Gumbel–Top-$k$) and affords fine-grained, lightweight control over a frozen sequential recommender backbone.

**Discussion on Surrogate Objective.** As mentioned before, our training objective is variationally inspired rather than a strict ELBO, because the forward pass uses hard k-hot Gumbel–Top-$k$ masking while the KL term assumes independent Bernoulli gates. This type of approximation is standard in sparse gating and masking methods, where discrete control variables are optimized using continuous relaxations or STE (Bengio et al., 2013; Maddison et al., 2016; Jang et al., 2016; Louizos et al., 2017), and the KL term in such frameworks primarily functions as a sparsity-inducing regularizer rather than an exact posterior-matching term. In our setting, the mask acts as a control signal rather than a latent probabilistic variable, and empirical behavior is far more critical than variational tightness. We observe stable optimization, smooth mask activations, and reliable request-conditioned effects across datasets and backbones. Thus, although our objective is not a strict ELBO, it follows well-established practices in sparse neural control and maintains the desired inductive bias while remaining computationally tractable.

# B. Details of Experiments

## B.1. Data Preprocessing

We preprocess all four datasets (ReDial[2], KuaiSAR[3], Beauty and CDs&Vinyl[4]) in a unified manner to construct RecBole-style atomic files. For each user, we sort interactions chronologically and build the request text by leveraging and concatenating information from the three prior interactions (title, content, category, search keyword, etc.) before the current timestamp; when no prior history exists, we optionally fall back to the current interaction. To obtain positive instances, we filter interactions according to dataset characteristics: for ReDial and KuaiSAR we keep only positive interactions, while for Beauty and CDs&Vinyl we retain ratings greater than or equal to 4.0. Limited by the computing budget, we perform random downsampling on KuaiSAR, Beauty and CDs&Vinyl to use only a portion of the data in these datasets. Finally, only items and interactions that pass these filters are retained to form the atomic `.inter` and `.item` files used in training and evaluation. Table 4 shows the statistics of the preprocessed datasets.

*Table 4.* Statistics of the processed datasets. "#Items" represents the total number of items, "#Inters" represents the total number of interactions (each one contains a request), and "Average Chars" represents the average number of characters of requests.

| Dataset | #Items | #Inters | Average Chars |
|---|---|---|---|
| ReDial | 5207 | 36460 | 112.84 |
| KuaiSAR | 174895 | 260243 | 124.63 |
| Beauty | 44977 | 122485 | 226.85 |
| CDs&Vinyl | 76368 | 141213 | 973.21 |

## B.2. Training of Backbones

To control the variance, in our experiments, the parameters of both backbones (SASRec and BERT4Rec) under each dataset are trained using the default settings of the RecBole library. For specific parameter settings, please refer to RecBole v1.2.1 [5] (Zhao et al., 2021) .

## B.3. Optimal Settings

We summarize in Table 5 the optimal hyperparameter settings used in our experiments for each backbone $\times$ dataset configuration. All hyperparameters were tuned within predefined search ranges, and the best configuration was selected based on validation NDCG@10. Below we briefly describe each hyperparameter and its search space:

- $\rho$: the drop ratio of $m$; searched over $\{0.05, 0.10, 0.15, 0.20, 0.25\}$.

- **PLM**: the pretrained language model used as request encoder, selected from $\{$BERT, RoBERTa, MiniLM, Modern-BERT$\}$.

- $\theta_M$: the part of the Transformer backbone subject to masking; chosen from $\{$FFNs, $W_O$, FFNs+$W_O\}$.

- $\phi_t$: the fine-tuning regime for the PLM; one of $\{$none (fully frozen), last (fine-tune the last layer), all (end-to-end tuning)$\}$.

- $\lambda_{\text{KL}}$: the weight of the KL regularization term; tuned in $\{0.1, 0.2, 0.3, 0.4, 0.5\}$.

- **shared**: whether gates are sampled once and shared across the entire batch (1) or sampled independently per instance (0) in the training phase.

- $\tau$ **(0.7 $\rightarrow$ 0.3)**: the temperature annealing schedule for Gumbel–Top-$k$ sampling; choose from $\{$linear, exponential, cosine$\}$ with a uniform start value of 0.7 and an end value of 0.3.

---

[2] https://redialdata.github.io/website/
[3] https://kuaisar.github.io/
[4] https://amazon-reviews-2023.github.io/
[5] https://recbole.io/docs/

*Table 5.* Optimal hyperparameter settings for each backbone × dataset configuration.

| Backbone | Dataset | $\rho$ | PLM | $\theta_M$ | $\phi_t$ | $\lambda_{\text{KL}}$ | shared | $\tau\ (0.7 \rightarrow 0.3)$ |
|---|---|---|---|---|---|---|---|---|
| SASRec | ReDial | 0.20 | RoBERTa | $W_O$ | last | 0.4 | 0 | cosine |
| | KuaiSAR | 0.05 | ModernBERT | FFNs+$W_O$ | all | 0.1 | 0 | cosine |
| | Beauty | 0.10 | RoBERTa | FFNs+$W_O$ | last | 0.2 | 0 | cosine |
| | CDs&Vinyl | 0.10 | ModernBERT | FFNs+$W_O$ | all | 0.1 | 0 | cosine |
| BERT4Rec | ReDial | 0.15 | RoBERTa | FFNs | last | 0.3 | 0 | cosine |
| | KuaiSAR | 0.05 | ModernBERT | FFNs+$W_O$ | all | 0.1 | 0 | cosine |
| | Beauty | 0.10 | RoBERTa | FFNs+$W_O$ | last | 0.2 | 0 | cosine |
| | CDs&Vinyl | 0.05 | ModernBERT | FFNs+$W_O$ | all | 0.2 | 0 | cosine |

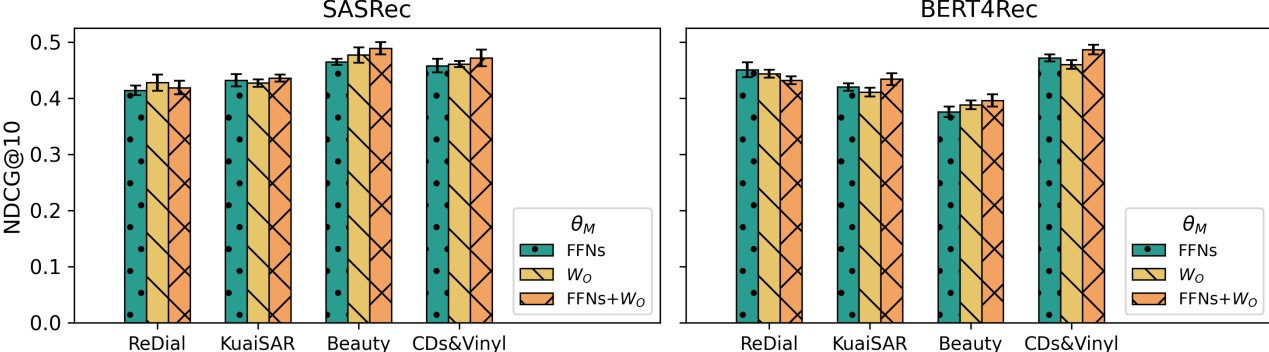

*Figure 4.* Impact of different scopes of $\theta_M$, evaluated using NDCG@10. For each setting, five evaluations were performed, the column height shows its average result.

## B.4. Detailed Experiments

In this section, we present more detailed experiments. In Section 4.3, we present experiments on sensitivity to $\rho$ and PLM selection, and here we present the other experiments.

**Scopes of $\theta_M$.** We further examine the impact of different masking scopes $\theta_M$ on model performance. As described in Section 3.3, we consider three options: masking only the feed-forward networks (FFNs), masking only the output projection of multi-head attention ($W_O$), and masking both jointly (FFNs+$W_O$). The results in Figure 4 show several consistent patterns. First, the overall performance across different scopes remains relatively stable, suggesting that CRAMER is robust to the precise choice of $\theta_M$. Second, the joint scope (FFNs+$W_O$) is most often optimal, particularly on larger datasets, where combining the two sources of control enables more expressive adaptation. Third, on smaller datasets, restricting the scope to either FFNs or $W_O$ alone can be advantageous, likely because a more constrained control space reduces the risk of overfitting when training data are limited (Bejani & Ghatee, 2021). This observation aligns with the intuition that FFNs mainly act as memory slots while $W_O$ governs attention aggregation–in low-data regimes, focusing on a single component may provide more stable and interpretable modulation, while in large-scale scenarios, the joint scope is more beneficial as abundant data supports richer request-conditioned adaptations and fully exploits the complementary roles of FFNs and $W_O$. In practice, a simple principle emerges: for large-scale, information-dense datasets, applying joint masking to both FFNs and $W_O$ is generally the best choice, whereas for smaller datasets, selecting either FFNs or $W_O$ individually may be preferable. These results confirm our design motivation in Section 3.3, where both FFNs and $W_O$ were identified as high-leverage control targets for request-aware adaptation.

**Regimes of $\phi_t$.** We further investigate the effect of different fine-tuning regimes for the request encoder parameters $\phi_t$, comparing three settings: (**none**) fully frozen PLM, (**last**) tuning only the last layer, and (**all**) end-to-end tuning. The results in Figure 5 show that CRAMER is generally robust across regimes, but some clear patterns emerge. Tuning only the last layer tends to yield stable gains over the frozen setting, particularly in smaller datasets or scenarios with limited request information, where modest adaptation is sufficient and helps avoid overfitting. In contrast, end-to-end fine-tuning becomes more beneficial in large-scale or information-rich datasets, where abundant data and longer request texts can support deeper adaptation of the PLM. However, full fine-tuning may occasionally harm performance in low-data regimes, reflecting optimization instability and overfitting risks. In practice, last-layer tuning offers a comparatively robust and reliable default, while full fine-tuning is best reserved for settings with sufficient scale and linguistic richness to fully exploit the request

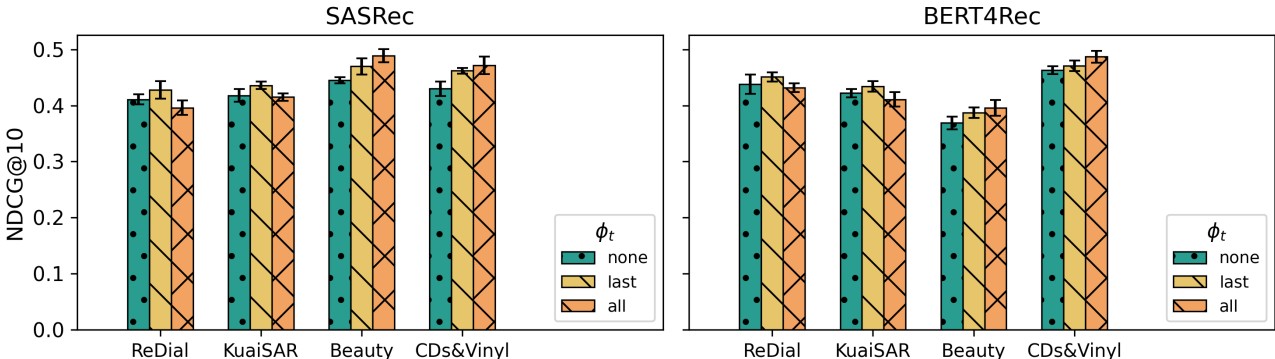

*Figure 5.* Impact of different regimes of $\phi_t$, evaluated using NDCG@10. For each setting, five evaluations were performed, the column height shows its average result.

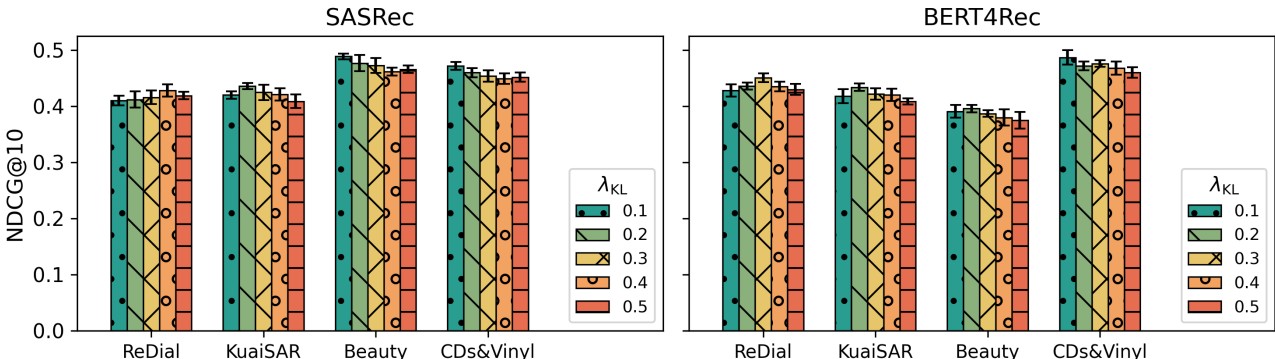

*Figure 6.* Sensitivity of CRAMER to weight $\lambda_{KL}$, evaluated using NDCG@10. For each setting, five evaluations were performed, the column height shows its average result.

encoder's capacity.

**Sensitivity to the Weight of KL Regularizer $\lambda_{KL}$.** We further study the influence of the KL regularization weight $\lambda_{KL}$ on model performance. Figure 6 reports NDCG@10 across different values of $\lambda_{KL} \in \{0.1, 0.2, 0.3, 0.4, 0.5\}$. Overall, the results indicate that CRAMER is relatively robust to the precise choice of $\lambda_{KL}$, with only moderate fluctuations across datasets and backbones. On smaller datasets such as ReDial, slightly larger values (around 0.3–0.4) tend to be more effective, likely because stronger regularization prevents overfitting under limited training signals. On the contrary, on relatively larger datasets such as KuaiSAR and CDs&Vinyl, weaker regularization (around 0.1-0.2) performs best, while overly large $\lambda_{KL}$ values consistently degrade performance by constraining the masks too strongly. Beauty shows an intermediate trend, where moderate values (around 0.2–0.3) strike a reasonable balance. These observations suggest a simple guideline: a small $\lambda_{KL}$ is generally sufficient for large-scale datasets, while moderate values are preferable for low-data regimes. Extreme settings should be avoided, as they either under-regularize or over-constrain the request-to-mask distribution.

**Masks Shared or Not.** We study whether the gating masks are sampled once and shared across the whole mini-batch (shared=1) or sampled independently for each instance (shared=0) during training. As shown in Figure 7, the non-shared regime dominates across all datasets and both backbones: its NDCG@10 is consistently and substantially higher. Notably, the best shared result never exceeds—and often trails well behind—the non-shared results. A plausible explanation is that sharing masks across an entire batch reduces the diversity of request-conditioned adaptation signals seen during training. This compromises the model's ability to align masks closely with individual requests, leading to systematically weaker representations. By contrast, sampling masks independently per instance maintains alignment between each request and its induced control signal, enabling more faithful request-to-mask adaptation and stronger predictive performance. From a training perspective, while the shared strategy can slightly reduce runtime overhead by avoiding per-instance sampling, this computational saving is outweighed by the clear and consistent performance degradation. Therefore, non-shared sampling should be regarded as the default choice, as it yields both more accurate and more reliable models in practice.

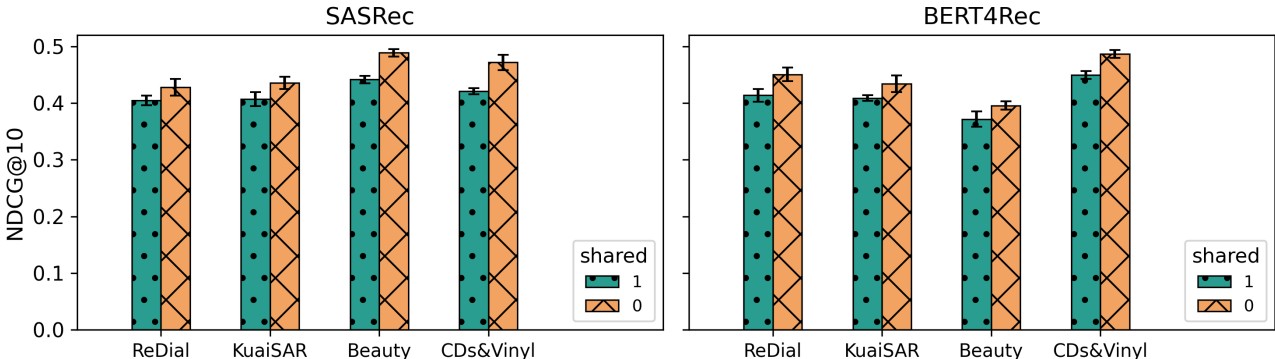

*Figure 7.* The effect of whether gates are sampled once and shared across the entire batch (1) or sampled independently per instance (0) in the training phase, evaluated using NDCG@10. For each setting, five evaluations were performed, the column height shows its average result.

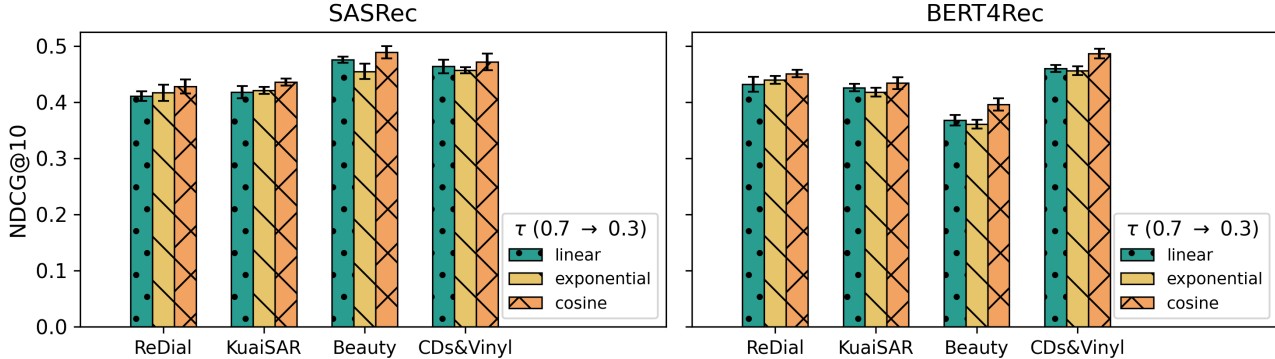

*Figure 8.* The effect of different temperature annealing schedule, evaluated using NDCG@10. For each setting, five evaluations were performed, the column height shows its average result.

**Temperature Annealing Schedule.** We further compare different schedules for annealing the Gumbel–Softmax temperature $\tau$ from 0.7 to 0.3 during training, including linear, exponential, and cosine decays. Figure 8 presents the results. Overall, cosine annealing achieves the best performance in most datasets and backbones, consistently outperforming linear decay and often surpassing exponential decay by a clear margin. Linear decay provides competitive results and is generally more stable than exponential, which tends to underperform due to overly rapid decreases in temperature at the early stages of training. The superiority of cosine annealing is likely because it offers a smoother and more gradual reduction, balancing exploration and exploitation more effectively while preserving sufficient stochasticity in the mask sampling process. In practice, cosine decay can be recommended as the default schedule, while linear decay remains a reasonable alternative when simplicity is preferred. Exponential decay is less favorable, as its aggressive early cooling can lead to suboptimal convergence and weaker final accuracy.

### B.5. Efficiency and Overhead (BERT4Rec)

*Table 6.* Inference efficiency comparison for BERT4Rec backbone. Runtime and GPU memory usage are measured as average per-request cost under identical settings. "Vanilla Backbone" reports the backbone-only cost, and the remaining rows show the additional overhead introduced by each request-aware method.

| Method | Runtime (s) | GPU Memory (MiB) |
|---|---|---|
| **BERT4Rec** | 0.038 | 2119.6 |
| +Query-SeqRec | +0.023 | +1620.4 |
| +BLaIR | +0.027 | +1102.6 |
| +LLM-ESR | +0.018 | +1375.3 |
| +REARANK | +9.184 | +9412.5 |
| +CRAMER (Ours) | +0.021 | +1408.1 |

*Table 7.* Controlled scaling study within the BERT4Rec family on KuaiSAR. Runtime and GPU memory report the vanilla backbone cost and the additional overhead introduced by CRAMER.

| Layers | Runtime (s) | +CRAMER (s) | Runtime Ratio | GPU Mem. (MiB) | +CRAMER (MiB) | Memory Ratio | NDCG@10 | +CRAMER |
|---|---|---|---|---|---|---|---|---|
| 4 | 0.038 | +0.021 | 55.3% | 2119 | +1408 | 66.4% | 0.366 | 0.434 |
| 6 | 0.056 | +0.026 | 46.4% | 2938 | +1543 | 52.5% | 0.379 | 0.447 |
| 8 | 0.075 | +0.032 | 42.7% | 3751 | +1712 | 45.6% | 0.376 | 0.451 |
| 10 | 0.094 | +0.039 | 41.5% | 4603 | +2011 | 43.7% | 0.385 | 0.462 |

## B.6. Controlled Scaling Study within BERT4Rec

To further examine how CRAMER scales with larger sequential recommender backbones, we conduct a controlled scaling study within the BERT4Rec family on KuaiSAR. Starting from the default 4-layer setting, we increase the backbone depth to 6, 8, and 10 layers while keeping the other settings unchanged. Table 7 reports NDCG@10 and the additional inference overhead introduced by CRAMER, together with runtime and memory ratios relative to the vanilla backbone.

The results show that CRAMER consistently improves recommendation performance across all tested backbone depths. Although the absolute runtime and memory overhead increase with deeper backbones, the relative overhead remains controlled and even decreases as the backbone becomes larger. This matches the row–column design: for each matrix $W^{(l)} \in \mathbb{R}^{\alpha_l \times \beta_l}$, entrywise modulation would require $\alpha_l \beta_l$ control variables, whereas CRAMER uses only $\alpha_l + \beta_l$ row–column gates. Thus, the control dimensionality grows linearly with layer width rather than quadratically with the parameter size.

## B.7. Further Discussion on PLM

To further examine whether CRAMER is inherently unstable with respect to the choice of request encoder, we conduct an additional study using four BERT-family PLMs with increasing capacity (Tiny, Mini, Medium, and Base) as the initialization of the request encoder (the "Base" version corresponds to the model used in the main experiments). As shown in Table 8, we evaluate CRAMER on SASRec across four datasets using NDCG@10.

All four datasets exhibit the exact same strictly monotonic ranking among the PLM versions (Tiny $<$ Mini $<$ Medium $<$ Base), which aligns precisely with their expected capacity ordering. For a single dataset, the probability of observing such a perfect ordering purely by chance is $1/24 \approx 0.0417 < 0.05$. Observing this pattern independently and consistently across all four datasets makes it highly unlikely that the variation originates from instability in CRAMER. Instead, these results indicate that CRAMER reliably leverages the semantic information provided by the request encoder: as PLM capacity improves, the encoder yields correspondingly richer representations of user requests, leading to more accurate and fine-grained control signals.

Importantly, this behavior does not represent a limitation of our framework, but rather reflects its inherently *forward-compatible* design. Even with classic and widely deployed PLMs such as BERT-base, CRAMER already achieves strong performance, and ongoing trends toward more capable yet efficient PLMs suggest that future lightweight encoders will only further enhance the model's ability to interpret user requests. Overall, the observed sensitivity arises primarily from intrinsic differences in PLM semantic expressiveness, and CRAMER remains both robust in practice and naturally aligned with continued advancements in text encoder architectures.

*Table 8.* Average NDCG@10 performance (five evaluations) of CRAMER under SASRec when initialized with four BERT versions of different capacity. Across all datasets, performance exhibits a strictly monotonic improvement that aligns with PLM capacity, illustrating that CRAMER faithfully leverages the semantic quality of the request encoder.

| BERT Version | Tiny | Mini | Medium | Base |
|---|---|---|---|---|
| ReDial | 0.403 | 0.411 | 0.419 | 0.428 |
| KuaiSAR | 0.420 | 0.425 | 0.432 | 0.436 |
| Beauty | 0.465 | 0.474 | 0.481 | 0.489 |
| CDs&Vinyl | 0.449 | 0.454 | 0.460 | 0.472 |

## B.8. Intuitive User Case Study

To complement the aggregate metrics, we further present an intuitive case study on five specific users from the CDs&Vinyl dataset. Table 9 lists the detailed information of these users. For every user, we compute the rank position of the ground-truth

*Table 9.* Detailed interaction information for the five users selected from the CDs&Vinyl dataset. For each user with a "User ID", "#Inters" represents the total number of this user's interactions, and "Last Item ID" represents the ID of the item in this user's last interaction. Because we use leave-one-out (LOO) evaluation, the item in the last interaction is the ground-truth item in evaluation.

| Index | User ID | #Inters | Last Item ID |
|---|---|---|---|
| #1 | AE25K5V5RESPJ4WKCALB3ZVYYQPQ | 11 | B000008KJ8 |
| #2 | AFE66HHU55NJMALT34HEODVGEPQA | 6 | B00KNTDO3I |
| #3 | AG2CJZJORAG7SG32SYNTNHICMGOQ | 8 | B07RF4JVGJ |
| #4 | AGUPFBZ756HTU4YIF4QKQEX3NS2Q | 13 | B00SFXFCWA |
| #5 | AHQF2VXWQPUBKYR3RMJ6VDFDYUSQ | 9 | B08L47S144 |

*Table 10.* Average ranking of true positive items of users selected from CDs&Vinyl (smaller is better). For each setting, five evaluations were performed, and boldface indicates the best, i.e., lowest, average rank under the same backbone.

| Method | #1 | #2 | #3 | #4 | #5 |
|---|---|---|---|---|---|
| **SASRec** | | | | | |
| \ | 18.2 | 28.4 | 27.0 | 19.0 | 17.8 |
| Query-SeqRec | 16.4 | 23.2 | 18.6 | 23.2 | 15.0 |
| BLaIR | 12.2 | 20.4 | 10.4 | 13.4 | 13.2 |
| LLM-ESR | 14.0 | 17.0 | 15.8 | 15.2 | 9.4 |
| REARANK | 13.6 | 23.2 | 11.2 | 10.0 | 11.4 |
| CRAMER (Ours) | **6.6** | **14.2** | **4.2** | **8.8** | **7.4** |
| **BERT4Rec** | | | | | |
| \ | 27.6 | 18.6 | 32.4 | 18.8 | 25.2 |
| Query-SeqRec | 21.2 | 17.2 | 14.0 | 17.0 | 21.2 |
| BLaIR | 19.4 | 11.4 | 16.2 | 13.0 | 14.2 |
| LLM-ESR | 11.4 | 13.8 | 18.2 | 7.6 | 13.8 |
| REARANK | 15.0 | 14.2 | 25.4 | 8.8 | 17.0 |
| CRAMER (Ours) | **7.0** | **8.2** | **11.4** | **5.2** | **12.0** |

positive item among all candidates under different backbones and request-aware methods. The results are summarized in Table 10.

As shown in Table 10, vanilla SASRec and BERT4Rec often place the ground-truth item at relatively low positions, indicating their limited ability to capture the user's immediate intent. Adding request-aware baselines consistently improves the ranking quality but still exhibiting instability and fluctuations across different users. In contrast, CRAMER achieves the highest ranks for all selected users under both backbones, demonstrating more reliable alignment with the user's natural-language request. This case study provides an intuitive, per-user confirmation that CRAMER delivers consistent improvements at the individual level beyond aggregate metrics.

### B.9. Details of Combination of Baselines and Backbones

In this section, we describe how each baseline is combined with the sequential recommendation backbones (SASRec and BERT4Rec) in our experiments. The combination strategies are detailed as follows:

**Query-SeqRec.** Query-SeqRec (He et al., 2022) integrates the request encoder with the sequential backbone. The backbone models the user's historical sequence, while the request encoder provides a semantic representation of the request. These two signals are fused through concatenation, and the fused representation is used to score candidate items.

**BLaIR.** BLaIR (Hou et al., 2024) encodes item metadata and natural-language requests into a unified embedding space, such that their representations can be directly compared. The cosine similarity between the semantic embedding and the item embedding is first computed in this shared space. Specifically, the semantic similarity score between the semantic embedding $v_q$ and the item embedding $v_i$ is first computed by the BLaIR encoder. This score is then integrated with the collaborative score from the backbone:

$$\text{score}(i) = \gamma \cdot \cos(v_q, v_i) + (1 - \gamma) \cdot f_\theta(s_u, i),$$

where $\gamma$ is a tunable fusion weight. In actual experiments, we set $\gamma$ to the optimal value on each backbonee × dataset.

**LLM-ESR.** LLM-ESR (Liu et al., 2024) augments the backbone with semantic embeddings derived from large language models. Specifically, each item $i$ is associated with both a semantic embedding $e_i^{\text{sem}}$ (pre-computed by an LLM and projected via an adapter) and a collaborative embedding $e_i^{\text{col}}$ (from the backbone). The user representation is similarly decomposed

into $(\boldsymbol{u}^{\text{sem}}, \boldsymbol{u}^{\text{col}})$. The final score is given by:

$$\text{score}(i) = [\boldsymbol{e}_i^{\text{sem}} : \boldsymbol{e}_i^{\text{col}}]^\top [\boldsymbol{u}^{\text{sem}} : \boldsymbol{u}^{\text{col}}].$$

**REARANK.** REARANK (Zhang et al., 2025) is used as a reranking stage on top of the backbone. The backbone first generates an initial ranking of candidate items based on the user's history. These candidates, together with the request, are then passed to the LLM reranker, which performs reasoning over the top candidates and outputs a refined ranking.

# C. Additional Clarifications and Analyses

This appendix provides additional clarifications, analyses, and implementation details that complement the main text. These materials are intended to improve clarity and reproducibility, and to make explicit certain design choices that are implicit in the main presentation.

## C.1. Clarification on the Variational Motivation

While Section 3.2 presents a variational lower bound formulation, we emphasize that CRAMER does not aim to perform exact variational inference over control signals. Instead, the variational perspective serves as a conceptual motivation for introducing structured sparsity and a KL regularizer that stabilizes request-conditioned gating.

In practice, the optimization objective in Equation (6) should be understood as a principled surrogate that balances predictive accuracy and control sparsity, rather than as a strict ELBO maximization. Similar uses of variational formulations as design guides have appeared in prior work on controllable generation and structured regularization.

## C.2. Non-Destructive and Reversible Masking

It is important to clarify that CRAMER's masking mechanism does not permanently modify or prune backbone parameters. All masks are applied multiplicatively at inference time and are conditioned on the current request. The underlying backbone parameters remain fixed and intact, and different requests induce different masks without interfering with each other.

This design enables reversible and non-destructive control: removing the request immediately restores the original backbone behavior, and no retraining or parameter update is required.

## C.3. Gradient Flow Through Discrete Gates

Although the gating vector $m$ is discrete, CRAMER employs a straight-through estimator (STE) with a continuous relaxation in the backward pass. This approach is a standard technique for training models with discrete latent variables and has been widely adopted in prior work.

Importantly, gradients never flow into the frozen sequential recommender backbone parameters; they are restricted to the request-to-mask module only. This ensures stable optimization while preserving the efficiency and modularity of the framework.

