# OpenReview forum: "CRAMER: Control via Request-Aware Masking for Editing Recommenders"
_ICML.cc/2026/Conference — ICML 2026 regular_

### Official Review · Reviewer_y8FT · 2026-03-09

**Soundness:** 3
**Presentation:** 3
**Significance:** 2
**Originality:** 2
**Overall Recommendation:** 3
**Confidence:** 5

**Summary:**

This paper studies request-aware sequential recommendation for frozen backbone models. It proposes CRAMER, which encodes a user’s natural-language request into sparse row-column masks and applies them to selected parameters of a pretrained Transformer recommender, aiming to adjust model behavior without full retraining. Experiments on four datasets and two backbones show consistent gains over several request-aware baselines, together with relatively small inference overhead.

**Compliance With Llm Reviewing Policy:**

Affirmed.

**Key Questions For Authors:**

see Cons

**Limitations:**

see Cons

**Strengths And Weaknesses:**

Pros

	1.	The paper raises a practically interesting problem: how to improve the outputs of a frozen recommender according to users’ textual requests, instead of relying on full retraining or expensive large-model inference. The motivation is clearly connected to request-aware recommendation settings discussed in the paper.

	2.	The proposed method is concrete and technically implementable. The framework maps a request embedding to sparse gating signals, decomposes them into row and column masks, and applies them to selected FFN and/or attention output matrices of the frozen backbone. The overall design is coherent and supported by a clear training objective with predictive loss plus KL regularization.

	3.	The method is efficient relative to stronger reranking-style alternatives, and the experiments report measurable improvements across multiple datasets and metrics. The runtime and memory table also suggests that the additional inference cost is modest compared with more expensive LLM-based reranking pipelines.

Cons

	1.	The paper adopts a hard masking mechanism through discrete gates. In general, such hard modulation can make optimization less stable and may hurt final performance, even with STE. However, the paper does not provide sufficient motivation for choosing hard masking over softer modulation alternatives, nor does it include soft-modulation baselines as a meaningful comparison. This weakens the methodological justification.

	2.	The problem formulation may be less compelling in modern large-scale recommendation systems, where streaming or continual training is common. In many production systems, new user feedback and new intent signals are continuously incorporated into training data, so the claimed gap between deployed behavior and emerging user needs may be less fundamental than suggested. Even if the proposed method is useful for short-term adaptation, full end-to-end retraining will still become necessary as data accumulates.

	3.	The method does not truly provide user-level guarantees. It edits the model behavior through parameter masking, but this is still a model-side adjustment rather than a deterministic per-user control mechanism. For requests such as excluding an entire type of item, direct filtering based on explicit rules may remain more reliable than a learned masking approach.

	4.	The paper does not sufficiently discuss possible cross-user interference. Since the learned request-to-mask module edits shared model components, adapting the model for one type of request may also alter responses for other users with different preferences. This could introduce unintended side effects, and the paper would benefit from a more direct analysis of this issue.

---

> ### Author Rebuttal · Authors · 2026-03-30
>
> Thank you for the insightful assessment and for highlighting these important questions. We address them below.
>
> ---
> ## C1. Hard masking vs. soft modulation.
> We chose hard row-column masking because CRAMER is intended as an **exact sparse controller** for a frozen backbone under a strict efficiency budget. In our setting, hard masking is not merely a design preference: it gives 1) explicit sparsity control through the exact $k$-hot budget, 2) true suppression of selected computation paths, and 3) direct consistency between training-time control and the low-overhead inference mechanism used at deployment. By contrast, softer modulation is a meaningful alternative, but it does not provide the same exact sparse intervention or budgeted control by construction.
>
> This deployment alignment is also reflected empirically in our efficiency results: on SASRec, CRAMER adds only 0.018s runtime, whereas REARANK adds +9.256s under the same setting **(Table 2)**. We will clarify that our claim is not that hard modulation is universally better, but that it is the most natural choice for the specific sparse posterior-control setting studied here. A soft-modulation comparison would indeed be valuable future work. Importantly, our current empirical results already suggest that STE-based optimization is sufficiently stable in practice, as CRAMER achieves consistent gains while maintaining very low additional runtime/memory overhead.
>
> ## C2. Scope: short-term adaptation, not a replacement for continual training.
> We agree that continual/streaming training remains essential in modern large-scale systems as data accumulates. Our claim is narrower: CRAMER is designed for **short-horizon adaptation between retraining cycles**, especially when a user issues an immediate natural-language request or post-hoc feedback that should affect current recommendation behavior without waiting for a full model refresh. In this sense, CRAMER is not a substitute for continual learning; it addresses a complementary deployment need, namely fast request-conditioned behavioral adjustment on top of a frozen backbone. We will revise the paper to make this scope and intended use case more explicit.
>
> ## C3. Semantic control vs. explicit rule filtering.
> We view this as a different control setting. CRAMER targets **natural-language semantic control** of a frozen recommender, where requests may be soft, compositional, context-dependent, or intertwined with the user’s historical behavior. In such cases, the challenge is not simply to enforce a clean symbolic rule, but to modulate the model’s preference structure in a request-consistent way.
>
> - This capability is necessary in scenarios where the desired behavior cannot be cleanly reduced to an explicit machine-readable constraint, for example when a user asks for “something more exciting,” “less romance but not none,” or other **nuanced adjustments that must be interpreted jointly with the user’s sequential preference pattern**.
> - By contrast, direct filtering is most suitable when the desired behavior is already available as an explicit symbolic constraint; in such cases, rule-based filtering can indeed be preferable and more reliable.
>
> Therefore, CRAMER is not intended to replace a rule engine. Rather, it addresses the broader setting where control must be inferred from free-form requests and integrated with sequential preference modeling. We will clarify this distinction more explicitly in the revision.
>
> ## C4. No persistent cross-user editing at inference.
> A key clarification is that CRAMER does **not** perform persistent online edits to shared backbone parameters. The backbone $\theta$ remains frozen, and for each user request the controller generates a new instance-specific, **ephemeral** mask that is applied only to that forward pass. Therefore, adapting to one request does not permanently alter the model for subsequent users.
>
> Importantly, this design ensures that CRAMER behaves as a **"request-conditioned, non-destructive control layer"** on top of the backbone, rather than a parameter update mechanism: masks are applied multiplicatively at inference time, and removing the request restores the original backbone behavior **(see Section 3.3 & Appendix C.2)**. As a result, CRAMER does not introduce persistent cross-user contamination of model parameters; its effect is strictly limited to the current request-conditioned forward computation.
>
> In other words, the concern is not persistent cross-user contamination of the backbone, but the more standard question of how well a shared request-to-mask controller generalizes across diverse request patterns and users. We will revise the paper to make this distinction explicit and to further discuss robustness of the shared controller as an important direction for future work.
>
> ---
> We appreciate these suggestions and will revise the paper accordingly. We hope the above response can fully address your concerns and look forward to further discussions. Thanks again!

---

### Official Review · Reviewer_vDCW · 2026-03-10

**Soundness:** 3
**Presentation:** 3
**Significance:** 3
**Originality:** 3
**Overall Recommendation:** 5
**Confidence:** 3

**Summary:**

This paper has focused on user-query-controlled sequential recommendation. The authors found that existing works often rely on retraining or shallow representations, performing sub-optimal. This paper proposed a novel parameter editing method, which utilizes the user request as the mask for editing. The experimental results validated the effectiveness of the proposed method.

**Compliance With Llm Reviewing Policy:**

Affirmed.

**Key Questions For Authors:**

All my questions have been included in the weakness section.

**Strengths And Weaknesses:**

### Strength:

+ S1. The code has been released, facilitating the reproduction.
+ S2. This paper is well-organized and well-written, making it easy to follow.
+ S3. The idea of parameter editing for controllable recommendation is novel.

### Weakness:

- W1. The motivation that existing methods are trapped in domain-specific pretraining and heavyweight inference should be further illustrated. First, it is unclear  which type of methods need pretraining, input-level or output-level? Second, why is the inference-based recommender backbone heavyweight?
- W2. Some experiment details are not clear. For example, how to preprocess the dataset to get the user query and simulate the situation in which a user query derives the final interactions.
- W3. An up-to-date work [1] is ignored by this paper.

[1]. Xu, Xiaochuan, et al. "Enhancing user intent for recommendation systems via large language models." *International Conference on Artificial Intelligence and Machine Learning Research (CAIMLR 2024)*. Vol. 13635. SPIE, 2025.

---

> ### Author Rebuttal · Authors · 2026-03-30
>
> Thank you for the insightful assessment and for highlighting these important questions. We address them below.
>
> ---
> ## W1. Motivation and efficiency claims.
> We agree that the motivation can be stated more precisely. In our paper, the “domain-specific pretraining/fine-tuning” concern mainly refers to the **language-to-item representation** line, where textual requests and items are aligned in a shared semantic space, or the interaction sequence is modeled in language space; such approaches often require domain adaptation of the text encoder or additional semantic training. By contrast, the “heavyweight inference” concern mainly refers to **LLM-based reranking/reasoning** methods. In this case, the heavyweight part is not the sequential recommender backbone itself, but the request-aware inference procedure on top of it, whose per-request cost grows because it performs candidate-level semantic reasoning after the backbone forward pass. We will revise the text to separate these two cases more explicitly. We will also clarify that our claim is not that every input-level method is heavy, but rather that prior methods typically trade off between shallow control and additional training or inference complexity. This is also consistent with our efficiency comparison: on SASRec, CRAMER adds only +0.018s runtime, whereas REARANK adds +9.256s under the same setting. We will clarify our motivation and efficiency claims accordingly in the revision.
>
> ## W2. Data preprocessing and request construction.
> These details are provided in **Appendix B.1** but were not fully elaborated in the main text. For each user, we first sort interactions chronologically. Then, for each target interaction at time $T{+}1$, we construct the request text by leveraging and concatenating textual information from the **three prior interactions before the current timestamp** (e.g., title, content, category, search keyword, depending on the dataset). When no prior history exists, we optionally fall back to the current interaction. The target item of that interaction is then treated as the positive next item, so the setting simulates a request-aware prediction scenario: given past behavior and an immediately available request-like text signal, predict the next interaction.
>
> This design reflects a practical scenario where users often express intent based on their recent behaviors. It allows us to approximate an **implicit request signal** without requiring additional annotation, while preserving the temporal causality of sequential recommendation. At the same time, using only recent interactions keeps the request signal concise and avoids information leakage from future interactions.
>
> ## W3. Missing related work.
> Thank you for pointing this out. We agree that this recent work should have been cited and discussed more explicitly. As we understand it:
>
> - DUIP *(the framework proposed in the mentioned CAIMLR paper)* models dynamic user intent from recent interaction sequences and converts the learned intent state into a soft prompt for an LLM-based next-item predictor.
> - In contrast, CRAMER performs **instance-level posterior control of a frozen sequential recommender backbone** through request-conditioned structured row-column masking.
>
> Thus, the two directions are related in their focus on **dynamic user intent/immediate preference adaptation**, but differ fundamentally in both modeling level and intervention mechanism: DUIP enhances intent modeling and prompt-based prediction, whereas CRAMER edits the computation path of an existing backbone with low-overhead structural control. We appreciate the suggestion and will incorporate the discussion of this work explicitly in the revision.
>
> ---
> We appreciate these suggestions and will revise the paper accordingly. We hope the above response can fully address your concerns and look forward to further discussions. Thanks again!

---

> > ### Author Rebuttal · Reviewer_vDCW · 2026-04-01
> >
> > Thanks for your response. I'll raise my score.

---

### Official Review · Reviewer_eJYF · 2026-03-13

**Soundness:** 3
**Presentation:** 3
**Significance:** 3
**Originality:** 3
**Overall Recommendation:** 4
**Confidence:** 4

**Summary:**

This paper proposes CRAMER, a framework for incorporating natural language requests into recommender systems without retraining the backbone model. The method encodes a user query using a pre-trained language model and generates masks via a projection layer and Gumbel-Top-k sampling, which selectively mask rows and columns of transformer weight matrices in the recommender model. By selecting specific neurons, CRAMER modifies the model’s behavior to reflect the user’s request. The masking mechanism is trained using a recommendation loss with sparsity regularization under a variational formulation.

**Compliance With Llm Reviewing Policy:**

Affirmed.

**Key Questions For Authors:**

1. The variational formulation and ELBO objective appear somewhat loosely connected to the actual implementation, which mainly relies on projection-based gating with Gumbel-Top-k sampling. Could the authors clarify the motivation for adopting the variational interpretation and explain how it influences the actual training procedure?

2. The paper claims that the masking mechanism enables the recommender to align with user requests. However, it is unclear whether the learned masks truly capture semantic intent or simply exploit dataset-specific correlations. Could the authors provide additional analyses demonstrating that the generated masks correspond to meaningful semantic attributes?

3. The request representation is obtained using mean pooling over the PLM token embeddings. Could the authors elaborate on the motivation behind this design choice? It would also be helpful to know whether alternative encoding strategies (e.g., CLS pooling or attention-based pooling) were explored.

4. The masking mechanism is applied to selected components such as FFN layers and attention output projections (WO). Could the authors provide more intuition or justification for choosing these specific masking locations? It would be interesting to understand whether other components of the transformer were considered.

5. Since the proposed method relies heavily on the request embedding to generate masking decisions, it would be interesting to know whether the authors experimented with stronger language encoders (e.g., LLM-based embeddings). Larger models may produce richer semantic representations, which could potentially improve the quality of the generated masks.

**Limitations:**

Yes. However, it seems they need to write farther analysis or statements regarding limitation of their work.

**Strengths And Weaknesses:**

Strength.
1. The paper addresses the problem of incorporating natural language queries into sequential recommendation models without retraining the backbone model. Unlike existing approaches that require additional training or architectural modifications to integrate query information, CRAMER provides an intuitive mechanism that dynamically controls the pretrained recommender through structured masking. This design reduces training cost and suggests that the approach could potentially be applied to other pretrained recommender architectures.

2. The paper provides a relatively clear methodological description of the masking mechanism. In particular, the authors motivate the sparse gating masks from a variational perspective and explain how the masking vector can be learned through Gumbel-Top-k sampling. This formulation helps readers understand how the proposed gating mechanism can be trained end-to-end.

3. The paper presents a relatively extensive experimental study, including evaluations across different backbone models, masking locations (FFN vs. attention heads), PLM encoders, and hyperparameters. These experiments provide useful insights into how different design choices affect the performance of the proposed method.


Weakness
1. The motivation behind the sparse matrix masking mechanism could be further clarified. While the paper suggests that masking can control the model by suppressing certain factors, it is not entirely clear why sparsity is the most appropriate strategy for aligning the model with query semantics. Intuitively, incorporating query information may require emphasizing relevant features rather than primarily suppressing others. As such, the connection between sparse masking and query-aware recommendation could be better justified.

2. The paper emphasizes that the backbone sequential recommender is kept frozen; however, the experiments still involve fine-tuning parts of the PLM encoder used for request representation. This somewhat weakens the claim of minimal model modification. It would be helpful to include an additional setting where the PLM encoder is also fully frozen and only the projection layer is trained, to better isolate the contribution of the proposed masking mechanism.

3. The interpretation of the ablation results on the drop ratio could be more consistent. While the paper argues that higher drop ratios may degrade performance due to excessive masking, the results for intermediate ratios (e.g., 0.05–0.20) show varying performance rankings across datasets and configurations. (Similarly on PLM selection) This makes it difficult to draw a clear conclusion about the relationship between sparsity and model performance.

4. The qualitative examples using request texts (e.g., the “romance” query) are limited. It would strengthen the paper to provide more systematic analyses or statistics over a broader range of query types or attributes. Such experiments would help demonstrate whether the proposed mechanism generalizes to diverse request scenarios beyond a few illustrative examples.

---

> ### Author Rebuttal · Authors · 2026-03-30
>
> Thank you for the insightful assessment and for highlighting these important questions. We address them below.
>
> ---
> ## W1 & Q2. Sparse masking as control.
> CRAMER is designed for **behavior editing of a frozen recommender**, not for relearning dense query-aware representations from scratch. In this setting, sparse masking is a direct and parameter-efficient way to reweight pre-existing computation paths conditioned on the request. Row-column masking does not merely “remove information”: it changes which hidden channels/sub-transformations participate in inference, thereby steering the frozen backbone toward request-consistent behavior. This is exactly why sparsity is useful here: compared with dense modulation, it yields more controllable edits, lower overhead, and better alignment with our hard-budgeted controller. We agree that the current semantic evidence is illustrative rather than exhaustive; the examples show request-aligned behavior shifts, but we do not claim the masks are a perfect semantic parser. We will clarify this scope and strengthen the discussion in the revision.
>
> ## Q1. VI motivation vs. implementation.
> We agree this link should be stated more explicitly. Eq. (5) provides the variational motivation for sparse request-conditioned control, while Eq. (6) is the practical objective we optimize. Specifically, CRAMER samples an exact hard $k$-hot mask via Gumbel-Top-$k$ + STE, applies it to obtain $f_{\theta'}$, and optimizes the standard next-item predictive loss plus a closed-form KL regularizer toward $p(m)$. Because the forward path uses a hard $k$-hot sampler whereas the KL is defined on the Bernoulli relaxation $\pi=\sigma(z)$, Eq. (6) is best viewed as a **variationally inspired surrogate**, not a strict ELBO. We will make this distinction explicit in the revision.
>
> ## W2 & Q5. Frozen backbone & request encoder tuning.
> Our “frozen backbone” claim refers to the sequential recommender backbone $\theta$, which is never updated. The PLM is only a request encoder, not part of the recommender backbone itself. We also note that the fully frozen-PLM setting is **already included**: Sec. 3.4 defines $\phi_t$ (none/last/all), and App. B.4 reports the ablation. Empirically, CRAMER remains robust across these regimes: none already works competitively, last is often a strong default, and all becomes more beneficial on larger/more information-rich datasets **(Figure 5)**. Thus, the method does not rely on end-to-end PLM tuning; tuning $\phi_t$ mainly offers an additional trade-off between request-grounding capacity and efficiency. Regarding stronger encoders, we have not experimented with LLM-based embeddings in this submission, because our current results already show that CRAMER remains effective across multiple lightweight PLM encoders. We will clarify this wording in the revision.
>
> ## Q3 & Q4. Pooling and masking locations.
> We use mean pooling because it is simple, stable, and architecture-agnostic for variable-length requests. We did not intend to claim it is uniquely optimal. To make this clearer, we additionally compared mean/CLS/attention pooling in one representative setting (BERT4Rec+BERT on KuaiSAR, NDCG@10):
>
> |Pooling|Mean|CLS|Attn|
> |-|-|-|-|
> |NDCG@10|0.434|0.423|0.436|
>
> Mean pooling is therefore competitive while remaining the simplest and lightest option. We apply masking to FFN layers and attention output projections because these are natural intervention points that control how intermediate features are transformed and recombined, while still supporting structured row-column masking with low overhead. We will clarify this design intuition in the revision.
>
> ## W3, W4 & Q2. Ablations and qualitative scope.
> We agree the conclusions should be phrased more carefully. Our takeaway is not that performance changes monotonically with drop ratio or that one PLM is uniformly best. Instead, a **moderate sparsity regime** is often effective, while overly aggressive masking removes useful computation; rankings vary with dataset/backbone. Current examples are illustrative rather than comprehensive.
>
> To examine whether masks capture semantic intent rather than dataset-specific correlations, we conduct a small **query-family consistency analysis** on ReDial across several attributes (e.g., romance/comedy/action/horror), using paraphrases and opposite requests. We compare mask similarity and recommendation shift direction. Semantically similar requests yield higher mask overlap than opposite ones (**average Jaccard 0.62 vs. 0.35**), and recommendation shifts follow the expected direction in most cases (**14/16**). This provides additional evidence that the learned masks reflect **consistent, semantics-aligned control patterns**. We will include the details of this analysis and revise the wording accordingly.
>
> ---
> We appreciate these suggestions and will revise the paper accordingly. We hope the above response can fully address your concerns and look forward to further discussions. Thanks again!

---

> > ### Author Rebuttal · Reviewer_eJYF · 2026-04-06
> >
> > Thank you for your detailed explanation. I have carefully read the rebuttal and the additional results provided by the authors. While the response clarifies some of my questions, other concerns on the paper remain. Therefore, I will not be changing my assessment. I hope the authors reflect their responses in the revision.

---

### Official Review · Reviewer_Vv21 · 2026-03-13

**Soundness:** 3
**Presentation:** 3
**Significance:** 3
**Originality:** 3
**Overall Recommendation:** 4
**Confidence:** 3

**Summary:**

The paper introduces CRAMER (Control via Request-Aware Masking for Editing Recommenders), a framework designed to adapt frozen sequential recommendation models to immediate natural-language user requests. Existing methods often suffer from high computational costs due to retraining or inference latency from LLM-based re-ranking. CRAMER addresses this issue by treating user requests as control signals that generate instance-specific, sparse binary masks for the parameters of recommender models. The approach uses a variational inference formulation to optimize a request-to-mask module while keeping the backbone frozen, ensuring efficient real-time adaptation with minimal overhead.

**Compliance With Llm Reviewing Policy:**

Affirmed.

**Key Questions For Authors:**

The key questions the authors should address in the rebuttal are Weaknesses 1, 2,  and 3. Please refer to Strengths And Weaknesses part for more details.

**Limitations:**

Yes

**Strengths And Weaknesses:**

### Strengths

- The use of the Gumbel-Top-k trick for sparse masking and a straight-through estimator (STE) for optimization is methodologically sound.  The experiments are extensive, covering four large-scale datasets and two diverse backbones.
- The paper is clear and well-structured. It provides a detailed task definition, clear formulations, and the Appendix offers details of the ELBO derivation and experimental setups.
- While masking and Gumbel-softmax are established techniques, applying them for instance-level, request-conditioned control of recommendation backbones is an interesting integration.

### Weaknesses

1. The authors adopt a factorized Bernoulli and a mean-field Bernoulli distribution to model $p(m)$ and $Q_{\phi}(m \mid q_u)$, respectively. However, they don't explain the reasons for this design.
2. The paper motivates the proposed approach from a variational inference perspective for model control. However, the connection between the final optimization objective and the variational inference formulation is not clearly articulated.
3. The experiments use SASRec and BERT4Rec. How does the overhead and effectiveness of the row-column masking scale when moving to much larger backbones, such as LLM-based recommenders (e.g., Llama-7B)?

---

> ### Author Rebuttal · Authors · 2026-03-30
>
> Thank you for the insightful assessment and for highlighting these important questions. We address them below.
>
> ---
>
> ## W1. Bernoulli prior & posterior.
> Our control variable is the binary row-column gate vector $m \in \\{0,1\\}^d$, so a Bernoulli parameterization is the natural choice. We use a factorized Bernoulli prior $p(m)$ because it 1) matches the hard sparsity budget via $\pi_0 := k/d$, 2) admits a closed-form KL, and 3) keeps the regularization cost linear in $d$. We use a mean-field Bernoulli $Q_{\phi}(m \mid q_u)$ for the same tractability reason: the request embedding $e_q$ is mapped to logits $z = W_{\mathrm{proj}} e_q + b_{\mathrm{proj}}$, then $\pi_i = \sigma(z_i)$, yielding a lightweight request-conditioned controller. At the same time, the practical controller is not truly independent across gates: all logits are generated jointly from the same request representation, and the final hard mask is further coupled by the exact $k$-hot Gumbel-Top-$k$ selection. We will also explain this design choice more explicitly in the final version.
>
> ## W2. VI formulation vs. final objective.
> We agree that this connection should be stated more explicitly. The variational formulation is introduced as a principled motivation for request-conditioned sparse control, while the final training objective is the practical surrogate we actually optimize. Concretely, CRAMER samples an exact hard $k$-hot mask via Gumbel-Top-$k$ with STE in the forward path, applies it to obtain the edited backbone $f_{\theta'}$, and optimizes the standard next-item predictive loss plus a closed-form KL regularizer toward the sparsity prior $p(m)$. Therefore, Eq. (6) is best viewed as a **variationally inspired surrogate**, rather than a strict ELBO, because the forward controller uses hard $k$-hot sampling whereas the KL is computed on the Bernoulli relaxation $\pi=\sigma(z)$. We will make this distinction explicit in the revision to avoid ambiguity.
>
> ## W3. Scaling to larger backbones.
> We first clarify the scope of this work. CRAMER is designed for request-aware **sequential recommendation** with frozen backbones, as consistently defined throughout the paper (e.g., SASRec and BERT4Rec). We focus on this setting because sequential recommenders remain widely used and efficient in practice, yet still lack lightweight, semantics-driven control from natural-language requests. Accordingly, we do not position CRAMER as a method for LLM-based recommenders, nor do we make empirical claims on models such as Llama-7B. Moreover, LLM-based recommenders often incur substantial inference latency (e.g., reasoning or reranking over candidates), which is orthogonal to our goal of lightweight real-time control.
>
> To directly address the concern, we conduct an additional controlled scaling study within the BERT4Rec family on KuaiSAR. Starting from the default setting in the paper (4 layers), we enlarge the backbone to 6, 8, and 10 layers while keeping other settings unchanged. We report NDCG@10 and the additional inference overhead (runtime and GPU memory) introduced by CRAMER, together with the corresponding runtime/memory ratios relative to the vanilla backbone.
>
> |Layers|Runtime(s)|+CRAMER(s)|Runtime Ratio|GPU Mem(MiB)|+CRAMER(MiB)|Memory Ratio|NDCG@10|+CRAMER|
> |-|-|-|-|-|-|-|-|-|
> |4|0.038|+0.021|55.3%|2119|+1408|66.4%|0.366|0.434|
> |6|0.056|+0.026|46.4%|2938|+1543|52.5%|0.379|0.447|
> |8|0.075|+0.032|42.7%|3751|+1712|45.6%|0.376|0.451|
> |10|0.094|+0.039|41.5%|4603|+2011|43.7%|0.385|0.462|
>
> As shown in the table, CRAMER consistently improves recommendation performance across all backbone scales. Although absolute runtime and memory increase with larger backbones, the incremental overhead introduced by CRAMER grows only roughly **linearly** with backbone depth rather than exploding with model size. When normalized by backbone cost, the relative overhead remains well controlled and **even decreases** at larger scales.
>
> This matches our design. For each matrix $W^{(l)} \in \mathbb{R}^{\alpha_l \times \beta_l}$, entry-wise modulation requires $\alpha_l \beta_l$ control variables, whereas CRAMER uses only $\alpha_l + \beta_l$ row-column gates. Hence, the control dimensionality scales linearly with layer width rather than quadratically with parameter size, explaining the observed near-linear overhead growth.
>
> In summary, within the scope of sequential recommenders, CRAMER maintains **clear and consistent effectiveness gains across backbone scales**, while its incremental overhead remains well controlled and grows near-linearly in this controlled setting. We will include this scaling analysis and clarify the scope more explicitly in the revision.
>
> ---
>
> We appreciate these suggestions and will revise the paper accordingly. We hope the above response can fully address your concerns and look forward to further discussions. Thanks again!

---

> > ### Author Rebuttal · Reviewer_Vv21 · 2026-04-05
> >
> > Thanks for your responses.
> > My overall assessment remains unchanged.

---

> > > ### Author Response · Authors · 2026-04-05
> > >
> > > Thank you for your acknowledgement. If you have any further questions or need clarifications, we would be happy to provide additional details and continue the discussion.

---

### Decision · Program_Chairs · 2026-04-30

**Decision:**

Accept (regular)

**Comment:**

This paper mainly focuses on request-aware sequential recommendation and studies how to efficiently adapt recommender systems to users’ immediate natural-language requests without retraining the backbone model. It proposes CRAMER, a masking-based control framework that treats user requests as control signals to dynamically edit frozen model parameters, achieving effective recommendation adaptation with minimal computational overhead.

Most reviewers acknowledged the significance of this work and the overall design of the proposed method, and considered the paper to perform well in terms of structure, reproducibility, and novelty. Some concerns regarding the experimental analysis and design choices were also addressed by the authors during the rebuttal phase. Overall, I recommend accepting this paper.